# Identification of biomarkers in Alzheimer's disease and COVID-19 by bioinformatics combining single-cell data analysis and machine learning algorithms

**Juntu Li**[1☯], **Linfeng Tao**[1☯], **Yanyou Zhou**[1☯], **Yue Zhu**[2], **Chao Li**[1], **Yiyuan Pan**[1], **Ping Yao**[1]*, **Xuefeng Qian**[1]*, **Jun Liu**[1]*

**1** Department of Critical Care Medicine and Emergency, The Affiliated Suzhou Hospital of Nanjing Medical University (Suzhou Municipal Hospital), Gusu School, Nanjing Medical University, Suzhou Clinical Medical Center of Critical Care Medicine, Suzhou, Jiangsu, China, **2** Department of Breast and Thyroid Surgery, The Affiliated Suzhou Hospital of Nanjing Medical University (Suzhou Municipal Hospital), Gusu School, Nanjing Medical University, Suzhou, Jiangsu, China

☯ These authors contributed equally to this work.

* liujunphd@sina.cn (JL); qxfly3106@163.com (XQ); 13812797177@126.com (PY)

## Abstract

### Background

Since its emergence in 2019, COVID-19 has become a global epidemic. Several studies have suggested a link between Alzheimer's disease (AD) and COVID-19. However, there is little research into the mechanisms underlying these phenomena. Therefore, we conducted this study to identify key genes in COVID-19 associated with AD, and evaluate their correlation with immune cells characteristics and metabolic pathways.

### Methods

Transcriptome analyses were used to identify common biomolecular markers of AD and COVID-19. Differential expression analysis and weighted gene co-expression network analysis (WGCNA) were performed on gene chip datasets (GSE213313, GSE5281, and GSE63060) from AD and COVID-19 patients to identify genes associated with both conditions. Gene ontology (GO) enrichment analysis identified common molecular mechanisms. The core genes were identified using machine learning. Subsequently, we evaluated the relationship between these core genes and immune cells and metabolic pathways. Finally, our findings were validated through single-cell analysis.

### Results

The study identified 484 common differentially expressed genes (DEGs) by taking the intersection of genes between AD and COVID-19. The black module, containing 132 genes, showed the highest association between the two diseases according to WGCNA. GO enrichment analysis revealed that these genes mainly affect inflammation, cytokines, immune-related functions, and signaling pathways related to metal ions. Additionally, a

**Data Availability Statement:** Publicly available datasets were analyzed in this study. This data can be found here: https://www.ncbi.nlm.nih.gov/geo/

query/acc.cgi?acc=GSE213313, https://www.ncbi.
nlm.nih.gov/geo/query/acc.cgi?acc=GSE5281,
https://www.ncbi.nlm.nih.gov/geo/query/acc.cgi?
acc=GSE63060 and https://www.ncbi.nlm.nih.gov/
geo/query/acc.cgi?acc=GSE181279.

**Funding:** Jun Liu received the following awards:
Key Social Development Project of Jiangsu
Province (BE2021660), Key R&D Plan Projects in
Kunshan City (KSF202105), Suzhou Science and
Technology Project (SKY2023197) and Clinical
Research Project of Gusu School of Nanjing
Medical University (GSKY20240202). The funding
had no role in the study design, data collection,
data analysis, decision to publish, or preparation of
the manuscript. The research was conducted
independently by the research team.

**Competing interests:** The authors have declared
that no competing interests exist.

machine learning approach identified eight core genes. We identified links between these genes and immune cells and also found a association between EIF3H and oxidative phosphorylation.

## Conclusion

This study identifies shared genes, pathways, immune alterations, and metabolic changes potentially contributing to the pathogenesis of both COVID-19 and AD.

## Introduction

COVID-19, caused by the SARS-CoV-2 virus, primarily spreads through droplet and airborne transmission, resulting in respiratory symptoms and complications [1, 2]. This highly contagious virus has led to over 6.9 million deaths worldwide [3]. In some patients, infection triggers a cytokine storm, prompting an excessive immune response, systemic inflammation, and tissue damage [4, 5]. Due to its significant impact on morbidity and mortality, COVID-19 has become a critical concern in healthcare [6]. Timely detection of the SARS-CoV-2 virus is crucial [7]. The virus prompts an inflammatory response in the immune system [8]. Immune cells, such as macrophages, release pro-inflammatory cytokines, including IL-6 and IL-1β [9]. These cytokines activate and recruit other immune cells, thereby strengthening the immune response [10]. However, in some cases, an overactive inflammatory response can lead to immune damage and tissue inflammation [11]. The role of immune cells in determining the course, severity, and prognosis of COVID-19 is crucial [12–14].

Alzheimer's disease (AD) is a complex neurodegenerative disorder characterised by progressive cognitive decline and behavioural changes. Its pathological features include β-amyloid plaques and neurofibrillary tangles.

Due to its immune response mechanism, COVID-19 affects elderly and immunodeficient patients disproportionately [15–17]. Many patients frequently report neurological symptoms [18, 19], and studies have consistently linked viral infections to a heightened risk of neurodegenerative diseases [20–22]. Therefore, it is plausible to infer an association between COVID-19 and AD. The association between pre-existing AD pathology and immune mechanisms may be further strengthened by SARS-CoV-2 infection.

Research has demonstrated that susceptibility to SARS-CoV-2 infection in hosts results in cognitive decline, which is mediated by systemic inflammation [23]. Multiple Mendelian randomization studies have indicated a causal relationship between AD and COVID-19 [24, 25]. Some of the same mutated genes have also been found to co-exist in both diseases, such as double mutations in the apolipoprotein E (APOE4) allele [26–28]. Additionally, the binding of SARS-CoV-2 viral proteins to host mitochondrial proteins may inhibit oxidative phosphorylation. This process is closely associated with immune cells in COVID-19 patients, and the interaction shows a bias towards age and specific cell types. The pathological process of AD is closely linked to oxidative phosphorylation. Mitochondrial dysfunction and oxidative stress contribute to neuronal damage and worsening of Alzheimer's disease. Mitochondrial dysfunction and oxidative stress contribute to neuronal damage and worsening of AD. Therefore, the pathophysiological processes of both AD and COVID-19 are closely related to oxidative phosphorylation.

Recent studies have highlighted the role of genetic factors in determining the severity of AD and COVID-19 disease. The APOE4 allele is a recognised risk factor for AD [29] and has also

been associated with increased severity of COVID-19 [30]. In addition, genetic variants in immune response genes may also lead to different clinical manifestations in AD patients infected with COVID-19 [31, 32].

To investigate the shared genes and their functions between AD and COVID-19, we analyzed gene chip data from the GEO database. Genes associated with both diseases were identified using differential expression analysis and weighted gene co-expression network analysis (WGCNA) techniques. We then used machine learning methods to identify eight core genes. Additionally, we compared twenty-two immune cell subpopulations in healthy and patient samples using the cibersort method. To enhance comprehension of the link between COVID-19 and AD, we investigated the correlation between the core genes, immune cells, and metabolic pathways. Our results were additionally confirmed through single-cell analysis.

From a macro-level perspective, the global pandemic has had a profound effect on the distribution of healthcare resources. As Tsoulfas observes, the advent of the SARS-CoV-2 pandemic has compelled the healthcare system to adapt to a novel situation [33]. Our study, conducted at the genetic and immune level in conjunction with other diseases (AD), offers a comprehensive understanding of the pathogenesis of this challenging adaptation.

## Results

### Identification of DEGs

Based on the GEO database, we obtained datasets for Alzheimer's disease (AD) and COVID-19. Then we acquired and integrated two AD datasets. Prior to conducting a differential expression analysis, the validity of data integration was assessed using principal component analysis (PCA). The findings demonstrated that while samples initially clustered distinctly by tissue type, the corrected AD dataset led to a substantial reduction in segregation between tissues while preserving biological variation, thereby substantiating the reliability of data integration. The *limma* package was utilized to identify differentially expressed genes (DEGs) between the COVID-19 (n = 34) patients and healthy controls (n = 11). DEGs were defined using the criteria of a *p.value* < 0.05 and |logFC| > 0.585, resulting in the identification of a total of 3587 DEGs, including 1738 up-regulated genes and 1849 down-regulated genes. Thus, we obtained a sample of 232 Alzheimer's patients and 178 healthy controls. Then 4961 differentially expressed genes (DEGs) were identified, consisting of 1869 up-regulated genes and 1738 down-regulated genes. The up- and down-regulated DEGs of the two datasets were intersected separately, resulting in a total of 484 intersected DEGs. Among these, 199 were up-regulated and 285 were down-regulated (Fig 1A and 1B).

### Analysis of GO function

We performed Gene ontology (GO) enrichment analyses on the common up-regulated and common down-regulated genes obtained above, respectively. The results indicate that the up-regulated genes primarily focus on responses to viral and symbiotic interactions, signaling through cytokines and interferons, and regulation of immune and viral processes (Fig 1C). And the down-regulated genes are mainly enriched in large subunit of the ribosome and its cytosolic components (Fig 1D).

### WGCNA

The sample GSM119676 in the GSE5281 dataset was significantly outlier and was judged to be anomalous, so 140 was chosen as the threshold to remove this anomalous sample. After removing the outlier sample, the remaining samples were analyzed further. To achieve a scale-

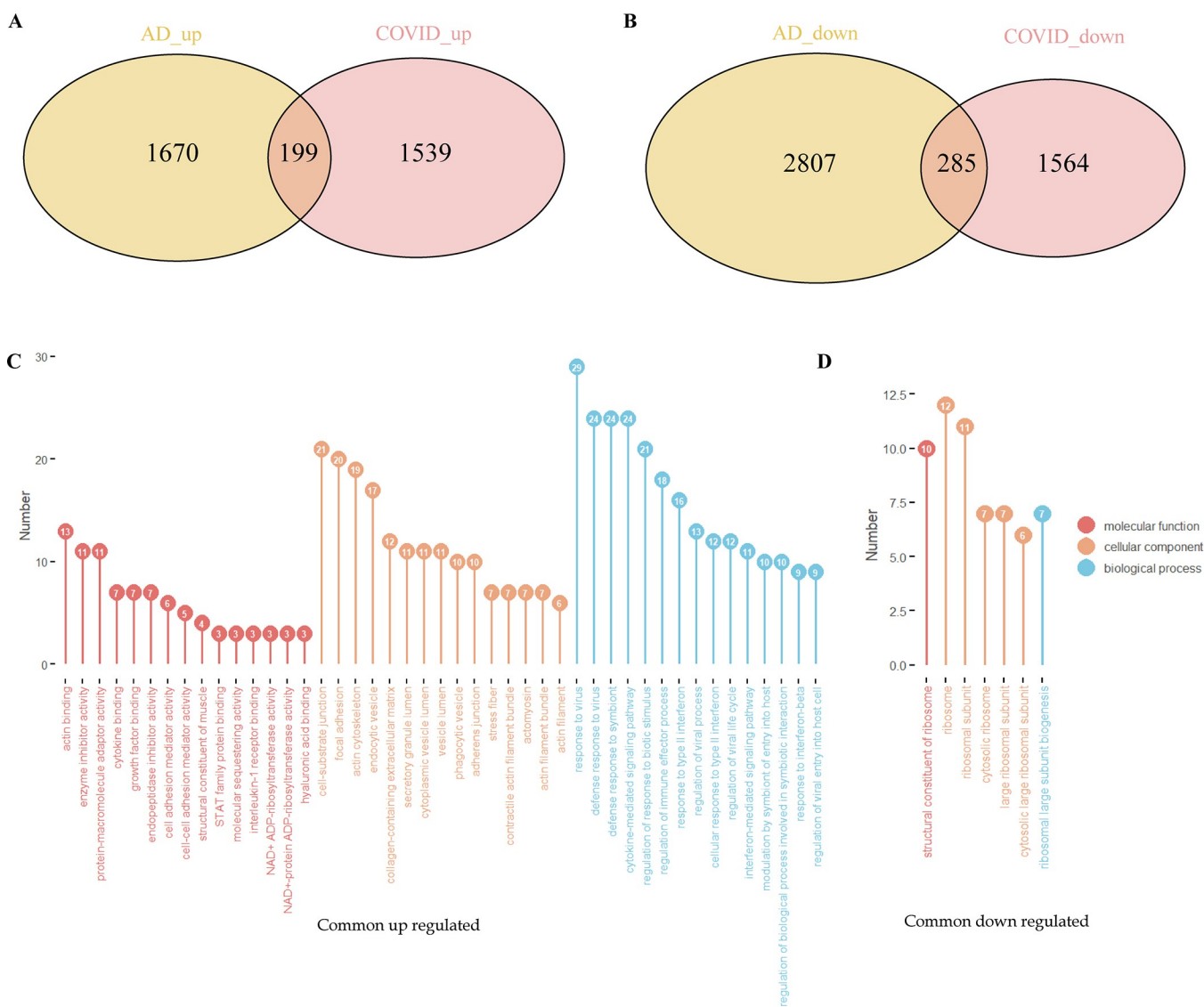

**Fig 1. Differential analysis and Gene ontology (GO) enrichment analysis of Alzheimer's disease (AD) and COVID-19 patients.** (A) Intersection of differentially expressed genes (DEGs) upregulated by AD and COVID-19 (B) Intersection of DEGs downregulated by AD and COVID-19 (C) GO enrichment analysis for common upregulated genes (D) GO enrichment analysis for common downregulated genes.

free network, the *pick Soft Threshold* function of *WGCNA* package was used to sift through power parameters ranging from 1 to 30, ultimately selecting a power of 6 as the soft threshold (S2C Fig). The identification of 9 modules, each comprising genes that share co-expression profiles, was facilitated by the *cuttree* dynamics and module signature gene function (S2D Fig). The threshold β was set at 6, and the TOM matrix was used to identify nine gene modules: blue (527), pink (435), red (172), yellow (113), black (132), brown (365), green (743), turquoise (7031), grey (128) (Fig 2A). The black module shows the correlation between AD and COVID-19 (Fig 2B and 2C). The genes in the black module have a positive correlation between the two diseases. The genes in the black module were analyzed for GO enrichment. The results revealed that these genes were involved in pathways related to cytokines, growth factors, apoptosis, and responses to metal ions and inorganic substances (Fig 2D).

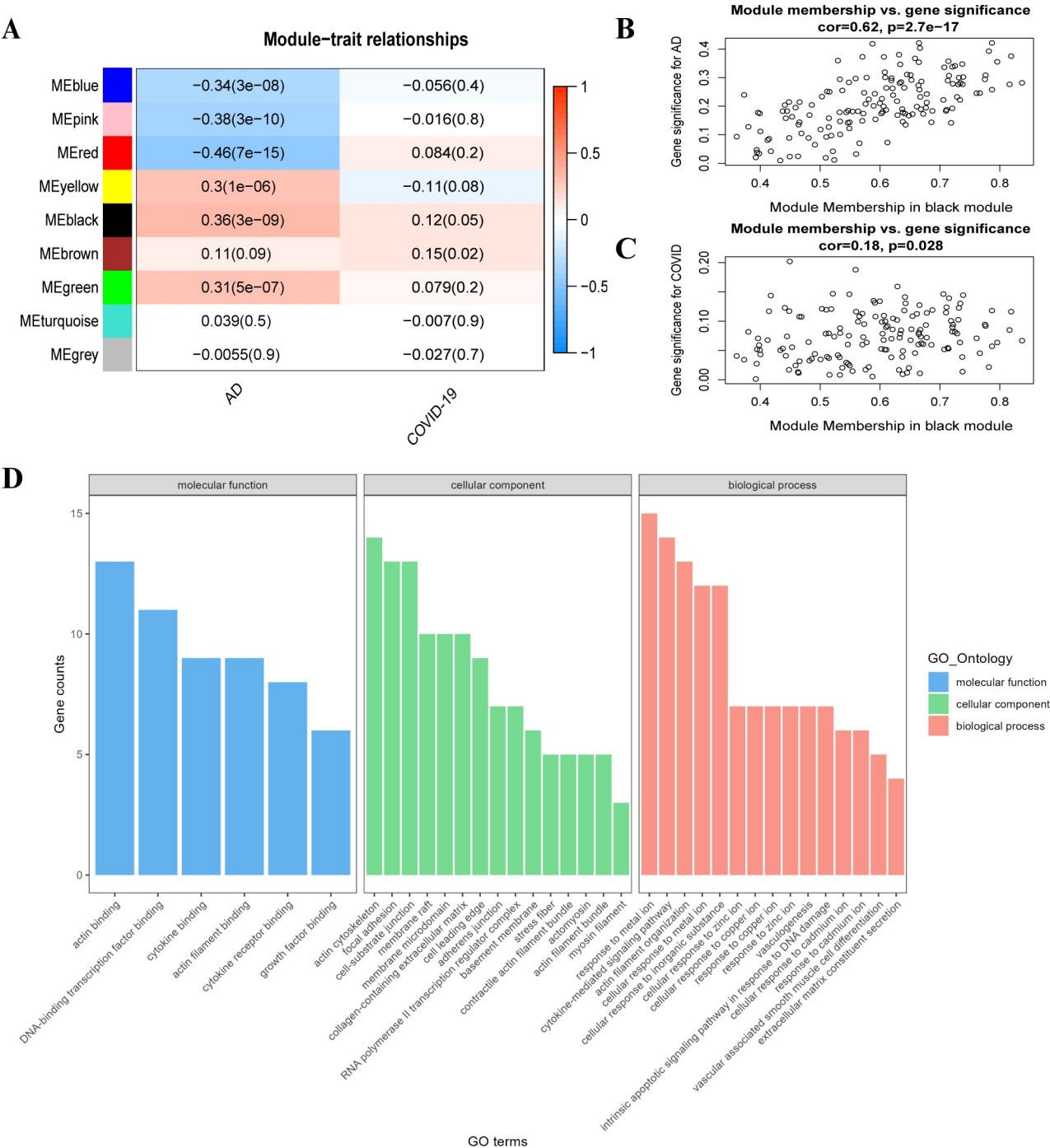

**Fig 2. Co-expression modules and enrichment analysis in patients with AD and COVID-19.** (A) The module–trait relationships in AD and COVID-19. Correlation analysis was performed using Pearson correlation within the WGCNA package. Correlation coefficients and corresponding p-values are provided for each module. (B) Correlation of black modules with AD (C) Correlation of black modules with COVID-19 (D) GO enrichment analysis for black module genes.

## Identification of central genes through machine learning

In order to identify the final set of core genes, we conducted a series of machine learning algorithm screenings. Initially, the relationship between the number of decision trees and the error

rate was evaluated (Fig 3A). It is evident that the error rate declines markedly with an increase in the number of decision trees. The error rate reaches a plateau at 200 decision trees. The augmentation in the number of decision trees resulted in a notable enhancement in the model's predictive precision. Subsequently, the gene importance score was calculated and the scores of 30 genes with an importance rating of greater than 0.7 were plotted (Fig 3B). The 22 genes with the highest importance scores (greater than 1) were then selected for inclusion in the add-down Least Absolute Shrinkage and Selection Operator (LASSO) regression algorithm for further analysis.

The impact of the regularization parameter λ on the regression coefficients was illustrated by the LASSO coefficient path diagram (Fig 3C). As the value of λ increased, the regression coefficients of an increasing number of genes converged to zero, indicating that these genes were removed from the model. To ascertain the optimal λ value, a cross-validation procedure was conducted. Based on the results, log(λ) = -3 was identified as the optimal λ value (Fig 3D). Subsequently, eight key genes were filtered out: ME3, SLC9A6, PCYOX1L, PRR11, GAS2L1, EIF3H, BCL6 and TTC19.

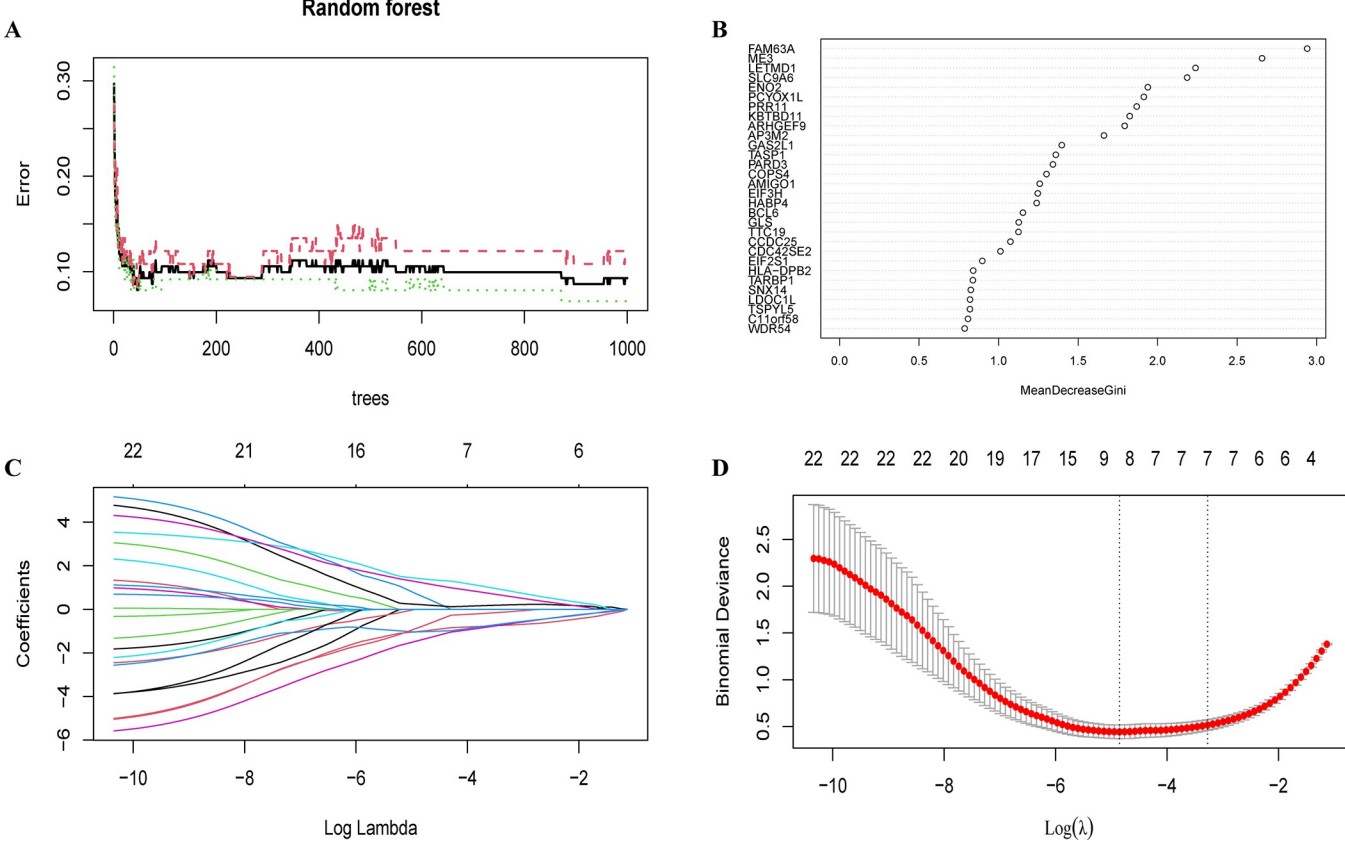

**Fig 3. Co-diagnostic gene screening and machine learning modelling.** (A) Indicates the relationship between the number of decision trees and error rate. The black solid, red, and green dashed lines indicate the change in error rate with the number of decision trees for overall, Alzheimer's, and COVID-19, respectively. (B) Plot of changes in gene importance scores based on the random forest algorithm. The horizontal axis indicates the importance of genes, and the vertical axis indicates genes with importance greater than 0.7. (C) Path diagram of Least Absolute Shrinkage and Selection Operator (LASSO) coefficients. The horizontal axis represents the logarithmic value of the regularisation parameter λ, and the vertical axis represents the value of the regression coefficient for each gene. Each curve represents the variation of the regression coefficient of a gene with λ. The numbers above indicate the number of non-zero coefficients at the corresponding λ values. (D) LASSO cross-validation curves. The horizontal axis indicates the logarithmic value of the regularisation parameter λ, and the vertical axis indicates the binomial deviation, which is used to measure the prediction error of the model. The red dots indicate the cross-validation error corresponding to each λ value, and the grey line indicates its standard error. The vertical dashed line on the left indicates the value of λ for the minimum deviation; the vertical dashed line on the right indicates the value of λ for the minimum deviation plus one standard error.

In order to make the screened genes truly valuable in the diagnosis and treatment of diseases, we further evaluated their diagnostic value by making receiver operating characteristic curves (ROC curves). The identified biomarkers show great potential for diagnosing AD and COVID-19. Additionally, these biomarkers offer insights into the interaction between the diseases, aiding in treatment strategies and drug target identification. This will also assist in the identification of drug targets and co-diagnosis of the diseases. The AUC values of the eight genes mentioned above were as follows: ME3 (0.898), SLC9A6 (0.850), PCYOX1L (0.918), PRR11 (0.850), GAS2L1 (0.855), EIF3H (0.892), BCL6 (0.824), and TTC19 (0.844). The AUC values of the obtained ROC curves were all greater than 0.8, indicating high accuracy and excellent predictive ability of the eight genes mentioned.

## Immune cell infiltration and immune cell correlation

We did an immune infiltration analysis of AD versus normal patients and constructed differential expression of immune cells with 8 central genes. In the same way, COVID-19 patients were analyzed. In comparison to normal controls, the AD patient group exhibited upregulation of CD4 naive T cells, regulatory T cells (Tregs), and resting Natural Killer cells (NK cells), while downregulating CD8 T cells and activated NK cells (Fig 4A). The expression of SLC9A6, EIF3H, and TTC19 showed a negative correlation with the infiltration level of neutrophils, Tregs, and Macrophages M0. BCL6, PRR11, and GAS2L1 were also found to be negatively correlated with CD8 T cells, CD4 memory T cells, memory B cells, and activated NK cells (Fig 4B). Similarly, CD4 naive T cells, monocytes, macrophages M0, activated mast cells and neutrophils were upregulated and CD8 T cells, CD4 memory resting T cells and resting mast cells were downregulated in the COVID-19 patient group (Fig 4C). The expression levels of SLC9A6, PCYOX1L, EIF3H, ME3, and TTC19 were found to be negatively correlated with the infiltration levels of activated mast cells, neutrophils, and macrophages M0. BCL6, PRR11, and GAS2L1 were negatively correlated with the infiltration levels of CD8 T cells, CD4 memory T cells, resting mast cells, and resting dendritic cells (Fig 4D).

To investigate the metabolic pathways of the central genes, we analyzed the correlations between eight central genes and classical metabolic pathways. The genes AD, BCL6, PRR11, and GAS2L1 showed significant positive correlation with hypoxia and significant negative correlation with fatty acid metabolism and oxidative phosphorylation (Fig 4E). SLC9A6, PCYOX1L, EIF3H, ME3, and TTC19 were found to have a significant positive correlation with fatty acid metabolism and oxidative phosphorylation, as well as a significant negative correlation with hypoxia. Similarly, for COVID-19, BCL6, PRR11, and GAS2L1 showed significant positive correlations with hypoxia, cholesterol homeostasis, xenobiotic metabolism, and glycolysis, and negative correlations with oxidative phosphorylation (Fig 4F). SLC9A6, PCYOX1L, EIF3H, ME3, and TTC19 were found to have a significant positive correlation with oxidative phosphorylation and a negative correlation with hypoxia, cholesterol homeostasis, xenobiotic metabolism, and glycolysis.

## Single-cell data analysis of AD patients

The single cell dataset underwent initial quality checks, assessing correlations among nFeature RNA, nCount RNA, and percent.mt (mitochondrial gene expression ratio) to confirm high-quality cell samples for the study. Fig 5A shows a positive correlation (correlation coefficient = 0.92) between nCount RNA and nFeature RNA, which represent unique molecular identifiers. Moreover, there is no significant correlation between both nCount RNA and nFeature RNA and percent.mt (correlation coefficients are -0.05 and -0.04, respectively). After excluding some cells, the results are presented in Fig 5B and 5C. The scRNA-seq dataset

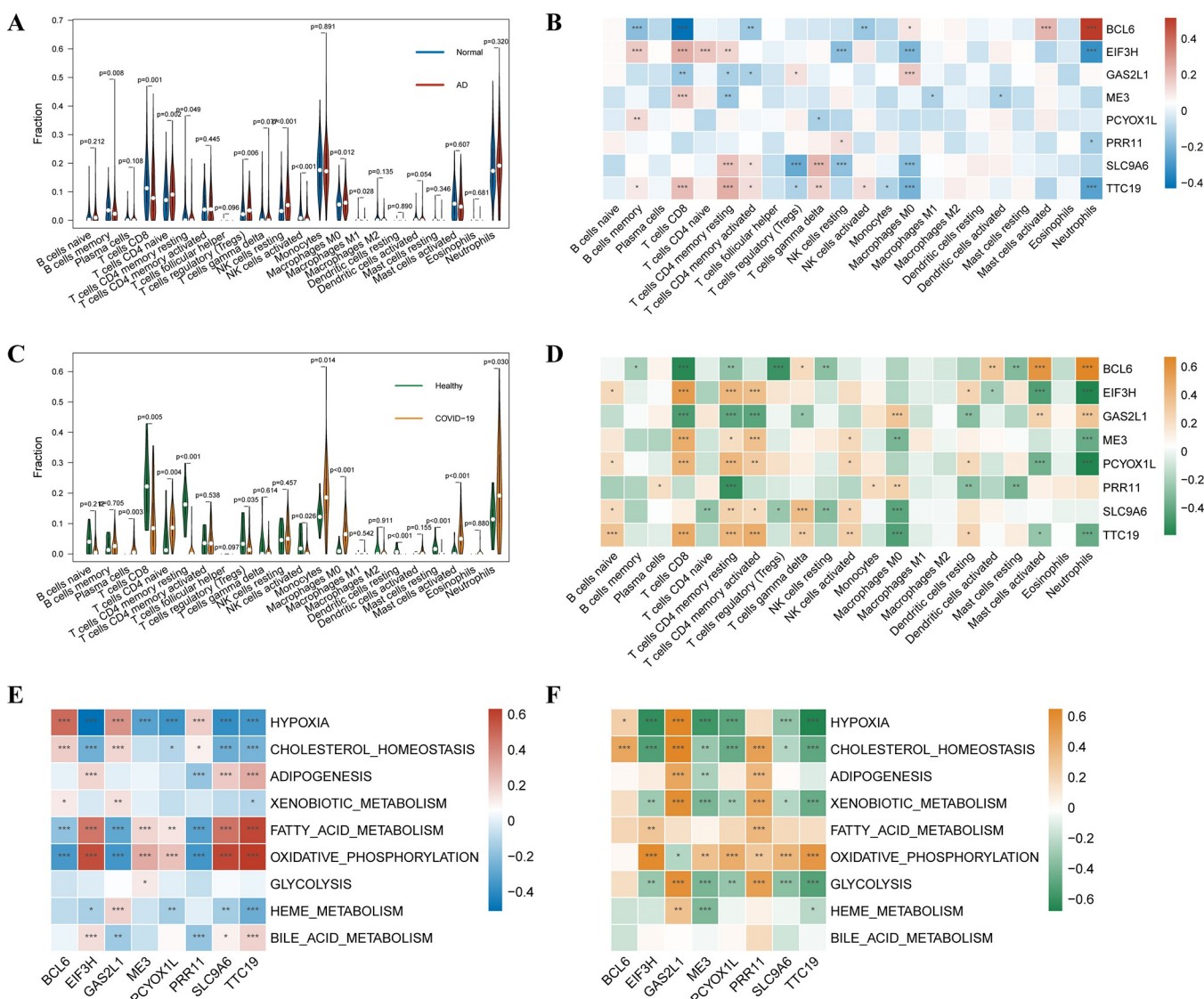

**Fig 4. Immune cells and metabolic pathways in patients with AD and COVID-19.** (A) Infiltration of immune cells between AD and healthy samples (B) Immune infiltration analysis of 8 candidate genes in AD (C) Infiltration of immune cells between COVID-19 and healthy samples (D) Immune infiltration analysis of 8 candidate genes in COVID-19 (E) Correlation between the expression levels of 8 hub genes and the ssGSEA enrichment scores for classical metabolic pathways in the AD data (F) Correlation between the expression levels of 8 hub genes and the ssGSEA enrichment scores for classical metabolic pathways in the COVID-19 data. *p < 0.05, **p < 0.01, ****p < 0.001. The heatmap colors represent correlation coefficients between central genes and immune cells (Panels B and D) or enrichment scores (Panels E and F). Red (for AD) and orange (for COVID-19) indicate positive correlations, whereas blue (for AD) and green (for COVID-19) indicate negative correlation.

revealed 3,000 genes exhibiting high variation levels. To further investigate these genes, we identified ten markers that stood out as particularly significant. A PCA on the top 20 PCs was performed (Fig 5D). Following this, the UMAP algorithms clustered cells into 4 distinct groups (Fig 6A). The AD group exhibited a decrease in NK cells and an increase in T cell subsets (Fig 6B). We therefore extracted mainly NK cells and T cells from the AD single cell dataset.

The expression levels of eight core genes were compared in normal subjects and AD patients. In both groups, EIF3H and TTC19 showed higher expression levels, with EIF3H being the most significant (Fig 6C). Violin plot display the proportions and the expressions of the eight core genes expressed in immune cells. EIF3H is expressed at a high level in all four

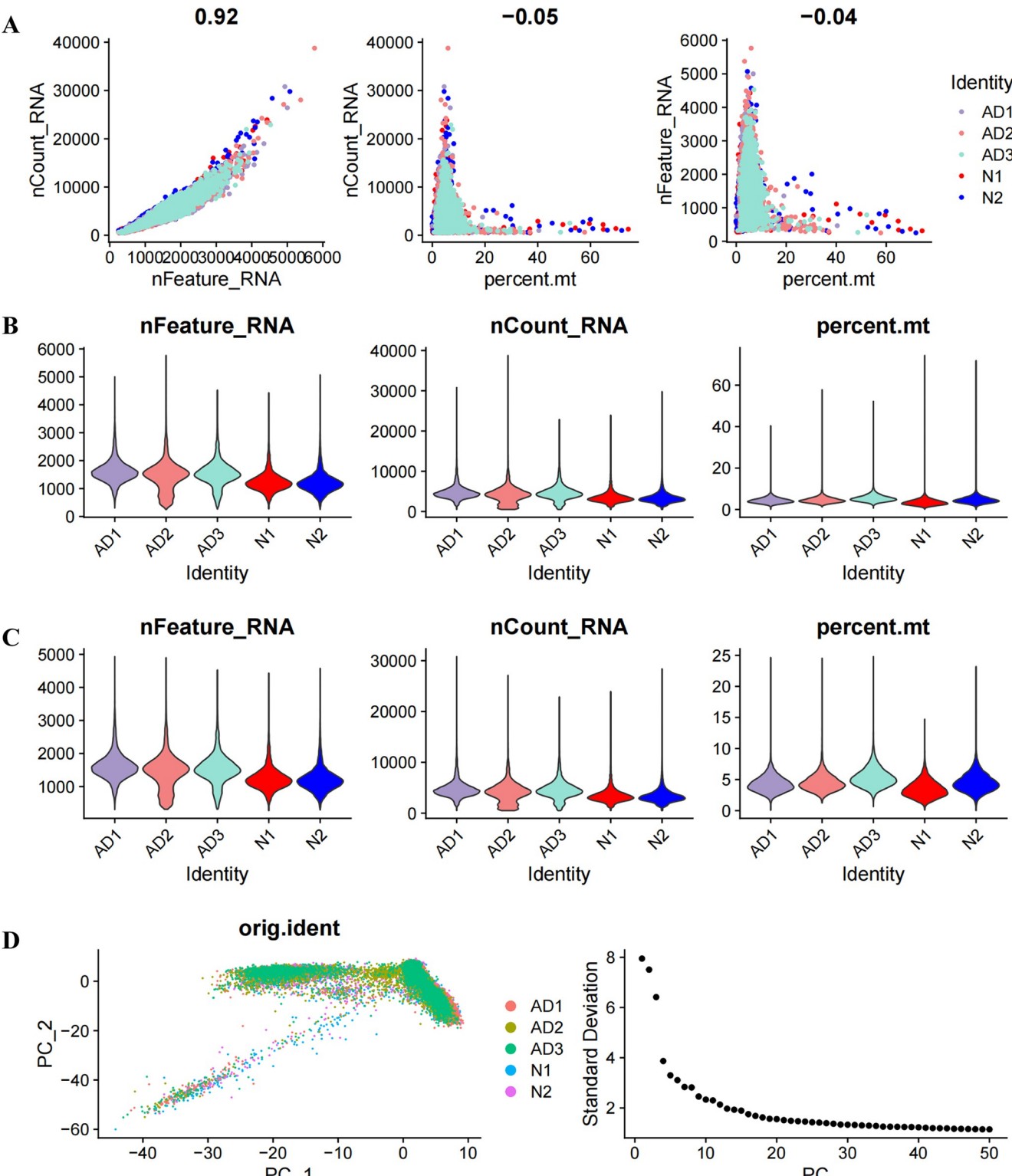

**Fig 5. Process for quality control of single-cell data.** (A) The relationship among gene expression, cell counts, and mitochondrial content within individual samples. The values at the top of each panel represent the correlation coefficients between the variables shown on the axes. (B) Percentage of mitochondrial genes (mt), RNA features (nFeatureRNA), and RNA counts (nCountRNA) for each sample prior to filtration. (C) Percentage of mitochondrial genes (mt), RNA features (nFeatureRNA), and RNA counts (nCountRNA) for each sample after filtration (D) Principal component analysis (PCA) plot, with each dot representing a cell, alongside an elbow plot utilized to ascertain the number of principal components (PCs).

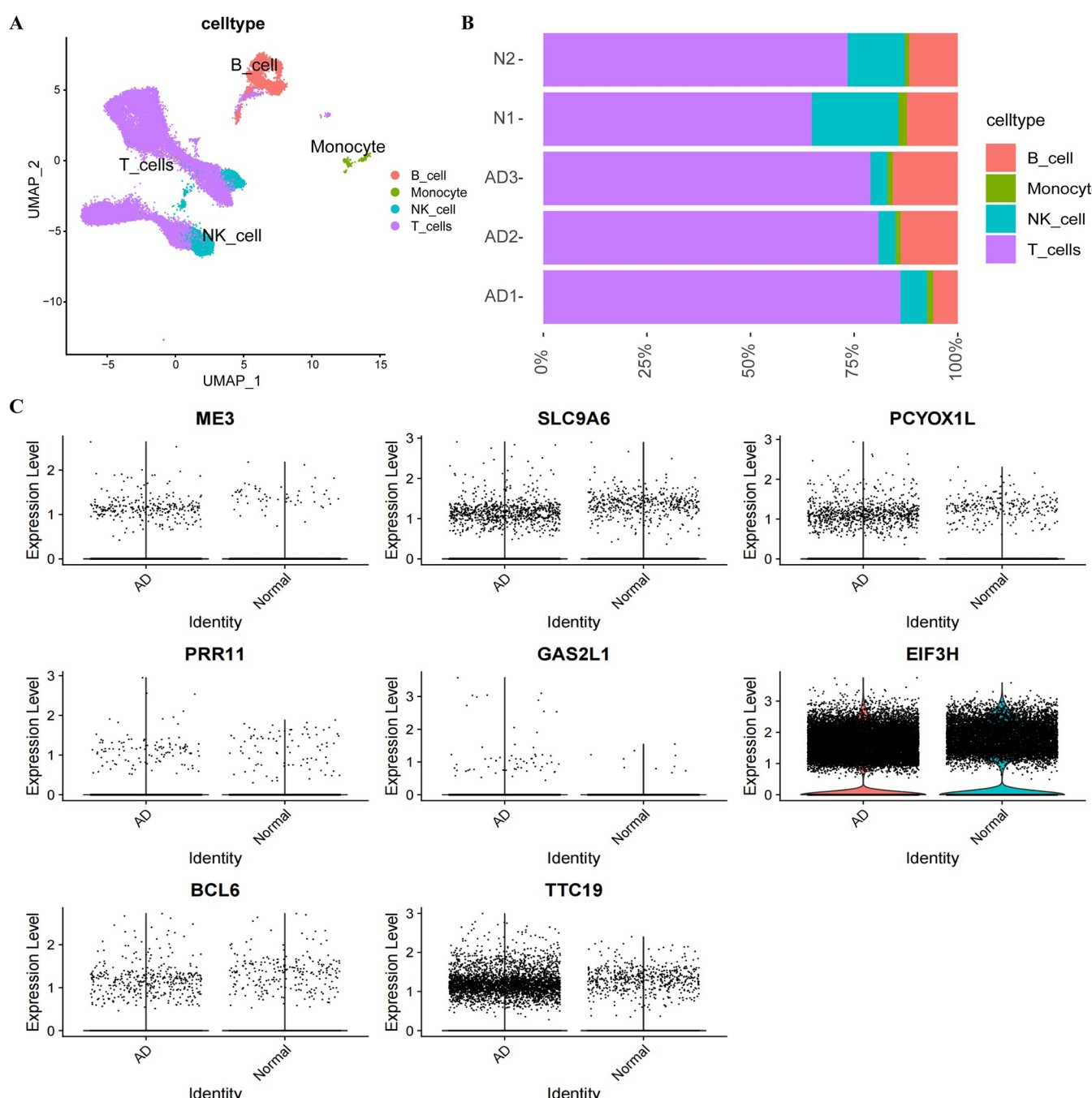

**Fig 6. Single-cell subpopulation identification and expression levels of genes in AD patients and normal controls.** (A) UMAP visualization illustrating cell subpopulations in patients with AD (B) Ratio of immune cell composition in AD patients to normal subjects (C) Comparison of expression levels of core genes.

immune cell types, both in terms of number and proportion, surpassing the expression levels of other genes (Fig 7A and 7B). Furthermore, we suggest co-localisation sites between the gene EIF3H and oxidative phosphorylation in immune cell subpopulations (Fig 7C). It is possible that EIF3H plays a role in the oxidative phosphorylation process of immune cells in AD patients.

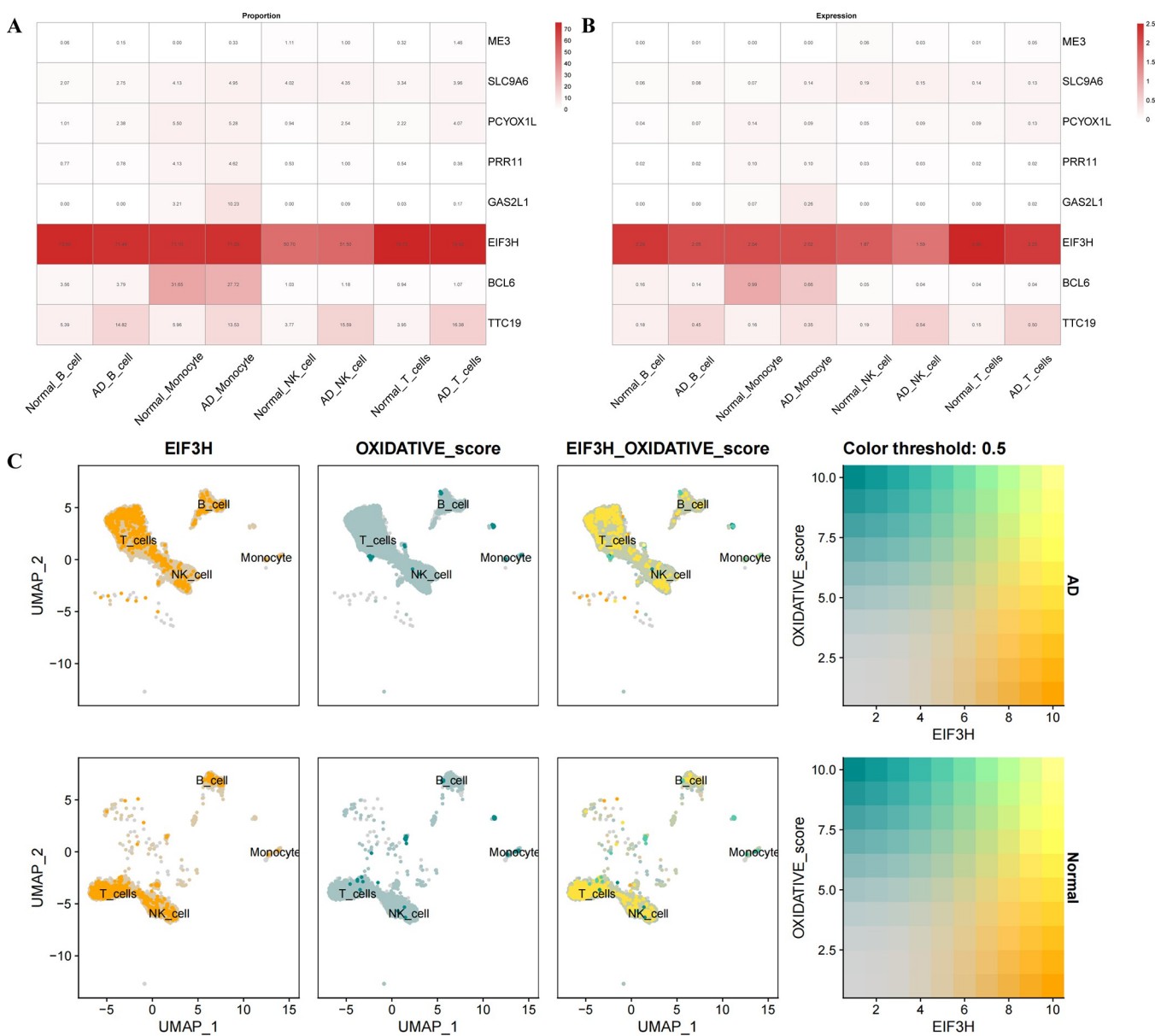

**Fig 7. Co-localisation and differential expression of core genes in immune cells of AD patients.** (A) Proportion of core gene expression in immune cells of AD patients and normal subjects (B) Core gene expression in immune cells of AD patients and normal subjects (C) Co-localisation of oxidative phosphorylation metabolic pathways and EIF3H in AD patients and healthy subjects.

## Discussion

In recent years, bioinformatics has rapidly developed, providing technical and methodological support for utilizing database information. This has made it possible to explore the pathophysiological links between Alzheimer's disease (AD) and COVID-19 using public databases. Infection with the SARS-CoV-2 virus triggers an immune response in the body [34, 35]. In some patients, this can lead to a cytokine storm, causing an excessive immune response [4, 5]. AD is characterized by dysregulation of the immune system, and the immune system is involved in the pathophysiological mechanisms of the disease [36–38]. As diseases closely related to the immune system, both AD and COVID-19 have been extensively studied. The association

between AD and COVID-19 is currently of interest due to compelling evidence from multiple studies supporting the link.

Research has demonstrated a significant correlation between genetic risk for AD and susceptibility to severe COVID-19, which is strongly associated with an inflammatory response [39]. Additionally, Mendelian randomization studies have suggested a causal relationship between AD and COVID-19 [25, 40]. According to a study, individuals with pre-existing cognitive impairment may be at a higher risk of SARS-CoV-2 virus infection and may experience a more severe prognosis after COVID-19 infection compared to those with normal cognitive function [41]. The combination of weighted gene co-expression network analysis (WGCNA) and machine learning approaches allowed us to identify key genes shared by AD and COVID-19. This facilitates the identification of patients in the early stages of the disease. Additionally, ssGSEA was used to assess the role of oxidative phosphorylation in the immunity of AD, further elucidating its inflammatory levels and pathological processes.

Numerous studies have explored the correlation between AD and COVID-19. However, there is a scarcity of comprehensive research on the shared genes, metabolic pathways, and immune cells between the two diseases. By utilizing WGCNA to identify common modules, we have discovered the core black module, which exhibits the highest degree of association with both diseases. Gene ontology (GO) enrichment analyses were employed to identify the principal pathways in which these genes were significantly enriched, with multiple enrichments into pathways related to cellular responses to metal ions, including copper and zinc ions. This is a topic worthy of further investigation.

The majority of metal ions are capable of binding to proteins, with approximately one-third of proteins containing metal ions [42, 43]. Metal ions play a pivotal role in protein structural stability and the regulation of biochemical reactions [44]. Additionally, metal ions act as cofactors in the maintenance of enzyme function and active catalysis [45, 46]. Notably, metal ions play a crucial role in the development, metabolism, redox reactions, and neurotransmitter transmission within the central nervous system [47]. Metal ions are deeply involved, not only in physiological processes of the organism, but also in pathological processes.

Metal ions have been demonstrated to exert control over the activation of immune cells through the regulation of proteases that mediate the inflammatory response [48]. The maturation and antigen presentation of dendritic cells is dependent on the involvement of zinc ions [49]. Furthermore, iron ions play a role in regulating the antimicrobial activity of macrophages and neutrophils [50]. Metal ions play a role in the onset and progression of numerous diseases, including those affecting the inflammatory response and the central nervous system [51, 52]. In autoimmune diseases such as rheumatoid arthritis, the involvement of zinc ions in metallothioneins has been demonstrated to exert a clear influence on immune cells [53, 54]. The abnormal regulation of proteases by metal ions has been demonstrated to promote and exacerbate inflammatory responses [55]. Metal ions also exert a profound influence on the progression of chronic inflammatory diseases such as hepatitis C, and studies have elucidated their potential value in therapeutic applications [56–58].

A substantial body of research has been conducted on the relationship between divalent metal ions, including copper, iron, and zinc ions, and infectious diseases prior to the emergence of the SARS-CoV-2 [59, 60]. These studies have significantly contributed to our understanding of pathogen-host interactions. In particular, the equilibrium of zinc and iron ions exerts an influence on the host immune system, including the functionality of T cells [61–63]. Recent studies have demonstrated the existence of metal ion imbalances in patients with COVID-19 [64], with evidence suggesting a role for copper ions in the progression of the disease [65].

Copper and iron have been demonstrated to influence the aggregation of amyloid-β (Aβ) [66–70], which represents a significant component of amyloid plaques observed in the brains of patients diagnosed with AD. Zinc also plays a role in Aβ aggregation, but has additional unique interactions due to its function in synaptic activity [71–73]. There is a strong association between tau protein hyperphosphorylation and iron ions [74, 75], which is another feature of AD. Copper and iron ions are implicated in redox reactions that give rise to oxidative stress, which is a pivotal factor in the exacerbation of cognitive dysfunction [74, 76–79]. Furthermore, our study suggests that metal ions, particularly copper and zinc, are associated with these two diseases. We hypothesize that COVID-19 influences the dynamics of metal ions, thereby affecting the progression of AD.

Although relevant articles have suggested that metal ions may play an important role in the neurological symptoms associated with the SARS-CoV-2 virus [47], there is still a paucity of research exploring the mechanisms through which the regulation of metal ion levels directly affects the progression of the disease in patients with AD. It is therefore crucial to conduct further research into the specific biological roles of metal ions and their potential therapeutic applications in the context of AD and the SARS-CoV-2 infection.

This study investigates the shared biomarkers between AD and COVID-19. Two machine learning techniques, random forest (RF) and Least Absolute Shrinkage and Selection Operator (LASSO), were used to select eight genes with co-diagnostic value: ME3, SLC9A6, PCYOX1L, PRR11, GAS2L1, EIF3H, BCL6, and TTC19. The effectiveness of these genes as diagnostic tools was validated by analyzing their receiver operating characteristic curves (ROC curves). Following the aforementioned identification, we were able to ascertain the high diagnostic potential of these biomarkers in AD and COVID-19. The construction of co-diagnostic tools can be facilitated by the use of the aforementioned biomarkers, thereby enabling the early and precise detection of these diseases. Molecular diagnostic methods are of significant value in the detection of SARS-CoV-2 infection. In particular, PCR-based techniques, which target viral genes such as ORF1ab, N and S proteins, are of particular utility in this regard [7]. The present study provides valuable additional insights into the host response during SARS-CoV-2 infection. Furthermore, this will facilitate an understanding of the interaction between the two diseases in these biomarkers, which will in turn provide new insights for the treatment of AD and COVID-19.

Most notable is BCL6. BCL6 has important regulatory roles in both Alzheimer's disease and COVID-19. In Alzheimer's disease, overexpression of BCL6 attenuates Aβ-induced neuronal damage and Tau protein hyperphosphorylation, thereby ameliorating neurodegenerative pathology [80]. And in COVID-19, BCL6 inhibits the hyperinflammatory response through modulation of the STAT signalling pathway, and its activity is associated with lung injury degree and immunomodulation, but persistently high expression may be associated with worse prognosis [81]. The involvement of BCL6 in both diseases through conserved molecular pathways such as STAT signalling suggests that alterations in its expression represent true disease mechanisms rather than tissue-specific effects. This suggests that BCL6 plays a central role in both neurodegenerative and severe inflammatory diseases by regulating key molecular networks, providing a potential target for the treatment of multiple complex diseases. In addition, the dual role of BCL6 in neurodegeneration and inflammation suggests that altered expression of this protein reflects disease-specific mechanisms instead of tissue-specific variants.

SLC9A6 is a gene that encodes a Na+/H+ exchange protein, which plays a pivotal role in maintaining intracellular acid-base balance and ion homeostasis [82]. It has been demonstrated that mutations in the SLC9A6 gene can result in a syndrome characterized by mental retardation, tau deposition and neurodevelopmental delay, as well as cognitive impairment

[83, 84]. These pathological changes and clinical manifestations are analogous to those observed in AD. The protein encoded by GAS2L1 is associated with the cytoskeleton and is involved in cellular morphology and movement [85, 86]. One of the most significant pathological characteristics of AD is the occurrence of morphological alterations in neurons, including the loss of synapses [87, 88]. The results demonstrated that SLC9A6 and GAS2L1 were also significantly differentially expressed in patients diagnosed with COVID-19. It is postulated that the alterations observed in patients with COVID-19, including oxidative stress and inflammation, may further exacerbate the dysfunction of these genes.

It is possible that mutations in ME3 and TTC19, both of which are associated with mitochondrial function, may affect mitochondrial function [89–91]. In particular, ME3 plays a significant role in mitochondrial energy metabolism. Mutations in TTC19 are involved in the pathological process of nerve injury [92]. Mitochondrial dysfunction represents a pivotal pathological mechanism in AD [93, 94]. Alterations in the expression of ME3 and TTC19 may impact neuronal survival and function by influencing mitochondrial energy metabolism and antioxidant capacity, thereby affecting neuronal survival and function. In patients with SARS-CoV-2 infection, inflammatory responses and oxidative stress may influence mitochondrial function by affecting the expression or function of ME3 and TTC19, which may in turn exacerbate neurological damage.

Furthermore, after conducting a thorough analysis of immune penetration and metabolic pathways, it was discovered that these diagnostic genes exhibited significant expression levels in various immune cell subtypes and metabolic pathways. We have found some interesting parallels between altered immune function in COVID-19 and other diseases. For example, a review has described analogous features of immune abnormalities between SARS-CoV-2 infection and chronic liver disease [95]. Dysregulation of the immune system in COVID-19 complicating various diseases seems to be an inextricable topic, and although it remains unclear whether the specific mechanisms are the same, our analyses provide new perspectives to understand the mechanisms of these immune dysfunctions. Notably, during the examination of single-cell data for AD, the EIF3H gene was observed to be highly expressed in multiple immune cell subtypes.

The gene EIF3H encodes a subunit of the eukaryotic translation initiation factor 3 (eIF3) complex [96]. This complex is crucial in the initiation phase of protein synthesis. EIF3H plays a significant role in facilitating the early steps of protein synthesis as part of one of the largest and most complex initiation factors [96]. Studies have demonstrated that EIF3H exhibits modified expression in certain cancer diseases, which correlates with oxidative phosphorylation [97]. Additionally, a possible association between eIF3h and oxidative stress has been suggested [98]. It has been shown in several studies that EIF3H is associated with viral infectious diseases, including rabies virus and hepatitis C virus (HCV) [99, 100]. Through metabolic analyses of AD and COVID-19, a relationship between core genes and oxidative phosphorylation has been identified. EIF3H was found to be strongly associated with oxidative phosphorylation in both diseases. Single-cell analysis of AD revealed co-localisation sites for EIF3H and oxidative phosphorylation within immune cells through co-localisation analysis. This corroborates the aforementioned linkage to some extent.

Furthermore, in light of our findings regarding metal ions and their established role in oxidative stress, we propose that these ions may also serve as key regulators of oxidative phosphorylation in both COVID-19 and AD. This regulatory role could be pivotal in elucidating the mechanisms of concurrence between SARS-CoV-2 infection and AD, particularly in the context of metal ion dysregulation.

During our preliminary analysis of AD and COVID-19, we observed potential correlations in the oxidative phosphorylation pathway among certain core genes. Our results suggest that

these genes may have a significant role in the development of these diseases. However, there is limited research on the gene EIF3H in relation to these diseases, and our findings should be considered preliminary. Further experiments are necessary to confirm our results.

Using WGCNA and machine learning methods, we screened 8 core genes. We then conducted immune infiltration analysis for two diseases and analyzed the relationship between the core genes, immune cells, and metabolic pathways. These findings have implications for the diagnosis of AD and COVID-19, and provide guidelines for further investigation into the link between these two diseases in terms of pathogenesis.

As time progresses, a significant number of patients who have contracted the SARS-CoV-2 virus recover [101]. However, this recovery is not without consequence, as it brings to the fore the issue of post-acute sequelae of SARS-CoV-2 infection (PASC). Some studies have demonstrated that SARS-CoV-2 infects and significantly affects the central nervous system (CNS), and that the long-term effects of SARS-CoV-2 on the organism can also cause an exacerbation of CNS symptoms [102, 103]. Some researchers have initiated studies to investigate the long-term effects of SARS-CoV-2 on the central nervous system (CNS). A sustained immune response to long-term consequences of SARS-CoV-2 infection has been observed, with notable effects on multiple organ systems, including the central nervous system [104]. Furthermore, studies have indicated that long-term sequelae of SARS-CoV-2 infection, also known as long-haul or post-acute sequelae of SARS-CoV-2 (PASC), are associated with mitochondrial dysfunction and oxidative stress [105]. In addition, there is evidence that metal ions such as iron ions are strongly associated with PASC [106]. Our results in this study fit with the above studies, which further suggests that metal ions and oxidative stress may play an important role in the pathological mechanism between COVID-19 and AD.

It should be noted, however, that the study is not without limitations. This study was primarily based on gene expression data, which lacked information on the clinical prognosis and mortality associated with the disease. This may restrict the scope of our analysis of the correlation between disease progression and the intensity of the immune response. Nevertheless, our investigation of gene expression and immune infiltration, and their interrelationship, offers significant insights into the pathogenic mechanisms and associations between genetic and immune aspects of AD and COVID-19. Further studies could investigate the combined influence of genes and specific immune responses (e.g., immune hyperactivation or immunosuppression) in AD and COVID-19, integrating clinical prognosis data, animal experiments, and other relevant information. Our future work will involve constructing animal models and using gene knockdown and other experimental methods to improve the study.

Furthermore, discrepancies in patient age, gender, geographical distribution, race and disease severity may influence the outcomes. These variables have the potential to influence the incidence and progression of numerous diseases, and the exclusion of the aforementioned variables may restrict the generalisability of our findings. The principal reason for the exclusion of these variables was the limited accessibility of public databases. The original objective of this study was to evaluate the contribution of biomarkers and immune cells, among other factors, to the pathophysiological processes of SARS-CoV-2 infection and AD. The absence of relevant variables may preclude further insights into the heterogeneity of the disease. Furthermore, the cross-tissue nature of the analysis of AD derived from brain tissue versus that of the analysis of COVID-19 derived from blood may potentially impact the accuracy of the results due to tissue-specific factors. In order to address this limitation, a comprehensive analysis was conducted, taking into consideration both methodological and biological aspects. Specifically, and firstly in terms of biological principles, it has been shown that changes in the histopathology of the brain in AD patients can be reflected by biomarkers in the blood [107]. The presence of common inflammatory markers in the CNS and blood has been demonstrated in studies of

AD and COVID-19 [108]. Both COVID-19 and AD may increase the permeability of the blood-brain barrier, suggesting that molecular exchanges between brain tissue and blood may exist in the context of both diseases [109, 110]. Secondly, in terms of data analysis strategy, every effort was made to eliminate tissue batch differences by the ComBat method. Moreover, the selected samples were subjected to gene chip technology, which ensured the consistency of the sequencing level and depth, thus guaranteeing the comparability between the samples. Finally, in terms of the results obtained, the black module was significantly associated with both diseases, despite the different tissue origins of the two diseases. The pathways we identified, such as metal ions and oxidative stress, play an important role in both diseases. These results suggest that the biological relevance far outweighs the tissue-specific effects. In conclusion, our study initially explored the mechanisms of association and potential therapeutic directions for COVID-19 and AD. It is recommended that future studies consider these factors in depth in order to gain a more comprehensive understanding of the subject matter. In the interim, we will continue to monitor the status of relevant available data with a view to conducting further relevant studies.

## Materials and methods

### Data acquisition and preprocessing

The gene expression data for Alzheimer's disease (AD) and COVID-19 was obtained from the GEO database (https://www.ncbi.nlm.nih.gov/geo/). The AD data were downloaded from the GEO database: GSE5281 from brain tissue (https://www.ncbi.nlm.nih.gov/geo/query/acc.cgi?acc=GSE5281) and GSE63060 from blood samples (https://www.ncbi.nlm.nih.gov/geo/query/acc.cgi?acc=GSE63060), while the COVID-19 data were obtained from blood samples in GSE213313 (https://www.ncbi.nlm.nih.gov/geo/query/acc.cgi?acc=GSE213313). In addition, we deleted 80 samples from the Mild Cognitive Impairment (MCI) patient group in dataset GSE63060. The *limma* package is a popular tool for gene discovery in differential expression analysis of microarray and high-throughput PCR data, with powerful read, normalisation and data exploration capabilities [111]. Therefore, the *limma* package was used to compare the normal and disease-affected groups. The screening criteria for differentially expressed genes (DEGs) were a *p.value* of less than 0.05 and an absolute logFC greater than 0.585. For single-cell correlation analysis, the GSE181279 data file with a total of 5 samples was downloaded.

We strategically integrated brain tissue and blood data for subsequent analyses of AD for the following reasons: Primarily, the direct reflection of pathological changes at the primary lesion sites of the disease by brain tissue sample data is of significance. Secondly, the systematic observation of pathological changes and potential biomarkers of the disease by blood sample data is noteworthy. Thirdly, the enhanced identification of molecular signatures shared between the central nervous system and peripheral tissues, facilitated by this combined analysis, is of considerable value in the identification of consistent pathological changes occurring between different tissues.

In order to ensure comparability of datasets across platforms, we implemented a rigorous data preprocessing procedure. Each dataset was first normalised independently using the normalizeBetweenArrays function in the *limma* package. Subsequently, the ComBat method was used to correct for batch effects between datasets. The effects of batch and tissue-specific variations were assessed by Principal Component Analysis (PCA), which confirmed the elimination of significant batch effects (S3 Fig) to ensure their consistency prior to downstream analyses. The corrected datasets showed no significant technical differences while maintaining biological differences.

## Enrichment analysis

Gene ontology (GO) enrichment analysis was performed using clusterProfiler package in R with org.Hs.eg.db annotation database. We obtained the biological functions and signalling pathways involved in disease occurrence and development. Enrichment was deemed statistically significant when the *p.value* < 0.01.

## Construction of co-expression networks and identification of hub modules

Weighted gene co-expression networks were constructed using the R package WGCNA (version 1.72–5) to identify genes associated with AD and COVID-19. Zero-expressed genes were filtered out before constructing the network to ensure data quality. Soft threshold power (β) was determined by analysing the scale-free topology model fit indices 1–20. Although the scale-free topology criterion ($R^2 > 0.9$) was initially reached at β = 5, we chose β = 6 because it provides better network stability while maintaining high network connectivity. Hierarchical clustering was performed using the flashClust algorithm to identify gene co-expression modules. Pearson correlation matrix was calculated to measure the co-expression similarity among the genes, which was then converted into adjacency matrix using the selected soft threshold power. Neighbour-joining matrices were further transformed into topological overlap matrices (TOM) to minimise the effect of noise and spurious associations. Modules were detected using a dynamic tree-cutting algorithm with the following parameters: minimum module size = 60 genes, deepSplit = 2 (default), and mergeCutHeight = 0.25.

## Feature selection by machine learning

Then, two machine learning algorithms, random forest (RF) [112] and Least Absolute Shrinkage and Selection Operator (LASSO) [113], were applied to identify key genes with high diagnostic potential. Random forest is advantageous for data processing of complex, advanced datasets. Using the R package *random forest*, we identified important genes. Genes with importance scores greater than 1.0 were selected. To efficiently screen the genes that contribute the most to disease diagnosis from the set of diagnostic genes selected by RF, we used Lasso regression as an additional screening tool. Blood samples from patients affected by both diseases were then utilized to assess the diagnostic value of the identified genes for both diseases in combination. This assessment was performed by plotting receiver operating characteristic (ROC) curves [114].

## Analysis of immune cell infiltration and immune cell correlation

The CIBERSORT algorithm was employed to evaluate the type and content of immune cells in the samples [115]. The CIBERSORT algorithm is a computational method developed to deconvolute the cell composition of complex tissues from their gene expression profiles. This algorithm incorporates 22 immune cell types, including T cells, B cells, monocytes, M1 and M2 macrophages, Natural Killer cells (NK cells), and others. The CIBERSORT algorithm was used to estimate the cell type in the sample using the genetic information obtained from the tissue sample. Specifically, we performed separate CIBERSORT analyses for the COVID-19 data (blood samples from GSE213313) and AD data (blood samples from GSE63060 and brain tissue samples from GSE5281). The analysis was conducted with 100 permutations and quantile normalization disabled (perm = 100, QN = F). And the relationship between immune cells and central genes in AD and COVID-19 was constructed.

Single-sample Gene Set Enrichment Analysis (ssGSEA) was performed to calculate the enrichment scores of gene sets in each sample. This method is based on the calculation of an

enrichment score for each sample-gene set combination. This is achieved by comparing the empirical cumulative distribution of genes in the gene set with that of all other genes.

## Single-cell data analysis

For the GSE181279 dataset, we utilized the *seurat* package for principal component analysis (PCA) and UMAP analyses, excluding cells based on feature and mitochondrial gene counts [116]. We normalized gene expression using *LogNormalize* and identified 3,000 HVGs with the *vst* method. We determined significant PCs through PCA and used the elbow method, then applied UMAP on 20 PCs. Cells were grouped into 27 clusters and DEGs were identified to ascertain cell types using *SingleR* with *HumanPrimaryCellAtlasData* as the reference [117, 118]. Finally, hub gene expression was showcased in immune cells using violin diagrams. Next, Pearson correlation was employed using the R package *stats* to explore the relationship between immune cells and metabolism, and co-localisation analyses between the important gene and metabolic pathways were performed.

## Statistical analysis

All statistical analyses were performed in the R language (Version 4.3.1). All statistical tests were considered statistically significant with a P value of $<0.05$.

## Conclusion

This research uncovers shared genetic markers, signaling mechanisms, immune system modifications, and alterations in metabolic pathways that may collectively contribute to the development of both COVID-19 and Alzheimer's disease (AD). By exploring these shared factors, researchers can gain a deeper understanding of the complex interplay between the two conditions. Additionally, identifying these shared factors creates new opportunities for diagnosing and treating both diseases. By focusing on these shared factors, researchers may potentially discover new methods to combat both COVID-19 and AD, ultimately enhancing patient outcomes and quality of life.

## Supporting information

**S1 Fig. Principal Component Analysis (PCA) plots of sample distributions from two AD datasets before and after batch correction.** (A) Prior to batch correction, samples from GSE5281 (brain tissue) and GSE63060 (blood) showed clear separation based on tissue origin. (B) After applying ComBat batch correction, there was a significant reduction in strict tissue-based separation, while maintaining biological variability.
(TIF)

**S2 Fig. Results of intermediate steps of data integration and WGCNA.** (A) PCA plots of datasets GSE213313, GSE5281 and GSE63060 before de-batching (B) PCA plot of datasets GSE213313, GSE5281 and GSE63060 after de-batching (C) Analysis of the network topology of soft threshold power (D) Cluster dendrogram identifying co-expressed genes (E) Hierarchical clustering before outlier removal (F) Hierarchical clustering after outlier removal (threshold: 140).
(TIF)

**S3 Fig. AUC curves for different genes.** (A) Gene BCL6 (B) Gene EIF3H (C) Gene GAS2L1 (D) Gene ME3 (E) Gene PCYOX1L (F) Gene PRR11 (G) Gene SLC9A6 (H) Gene TTC19.
(TIF)

## Author Contributions

**Conceptualization:** Linfeng Tao.

**Data curation:** Juntu Li, Linfeng Tao, Yanyou Zhou.

**Formal analysis:** Juntu Li, Linfeng Tao.

**Funding acquisition:** Jun Liu.

**Investigation:** Juntu Li, Ping Yao, Xuefeng Qian.

**Methodology:** Linfeng Tao, Yanyou Zhou, Yue Zhu, Ping Yao, Xuefeng Qian.

**Project administration:** Yanyou Zhou, Jun Liu.

**Resources:** Jun Liu.

**Supervision:** Yiyuan Pan, Ping Yao, Xuefeng Qian, Jun Liu.

**Validation:** Yue Zhu, Chao Li, Xuefeng Qian.

**Writing – original draft:** Juntu Li.

**Writing – review & editing:** Ping Yao, Jun Liu.

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
