## [Decision Letter · Decision Letter 0]

8 Oct 2024

PONE-D-24-36922Identification of biomarkers in Alzheimer's disease and COVID-19 by bioinformatics combining single-cell data analysis and machine learning algorithmsPLOS ONE

Dear Dr. Liu,

Thank you for submitting your manuscript to PLOS ONE. After careful consideration, we feel that it has merit but does not fully meet PLOS ONE’s publication criteria as it currently stands. Therefore, we invite you to submit a revised version of the manuscript that addresses the points raised during the review process.

We look forward to receiving your revised manuscript.

Kind regards,

Zhijie Xu

Academic Editor

PLOS ONE

Journal Requirements:

Additional Editor Comments:

The emerging roles of COVID-1 should be further discussed, like DOI: 10.2174/1573412918666220509032754, doi: 10.1016/j.drup.2023.100986, etc

Reviewers' comments:

Reviewer's Responses to Questions

**Comments to the Author**

1. Is the manuscript technically sound, and do the data support the conclusions?

Reviewer #1: Partly

Reviewer #2: Partly

Reviewer #3: No

2. Has the statistical analysis been performed appropriately and rigorously? 

Reviewer #1: I Don't Know

Reviewer #2: Yes

Reviewer #3: No

3. Have the authors made all data underlying the findings in their manuscript fully available?

Reviewer #1: No

Reviewer #2: Yes

Reviewer #3: Yes

4. Is the manuscript presented in an intelligible fashion and written in standard English?

Reviewer #1: Yes

Reviewer #2: Yes

Reviewer #3: Yes

5. Review Comments to the Author

Reviewer #1: Minor comments

- The authors should explain all the abbreviations, DEG for exemple…since the first use

- Figure 1, the authors need to precise for panel C &D “common up/down regulated” as maybe common between AD and COVID-19

- Figure 1 is written in small letter, as a reader we need to use the zoom in, maybe change the size of the police…

- Same remark for figure 2, we can not read without using zoom in

- Figure 2. The authors need to precise which correlation test they used.

- The authors need to rephrase “the results revealed that these genes were associated with pathways such as…” it is more likely that the pathways involving cytokines etc…

- Figure 5 A, the number on the top of the graph should be explained… represented r? p value ? or other?

- Figure 6 A and B, the writing on the panels are very small

Major comments

- The authors need to provide the GEO number of AD and COVID19 data set used, in the first sentence of the Materials and Methods/ Data download, the link refered to the principal webpage…and not in the results part

- The authors need to explain the limma package in short sentence…

- The authors should explain their choice for a threshold of 140 to remove the outlier samples.

- The authors need to explain why they used beta threshold at 6.

- The authors claimed in legend of the figure 7C co-localisation and in the text, association, maybe they need to explain these 2 concepts…

Reviewer #2: The work provides a interesting descriptive statistical analysis of comparing genetic regulation in Alzheimer’s disease and covid-19 patients. The presented statistical pipeline leads into a robust model that could be used for exploring relations between inherited and infectious diseases. The work shows a novel approach and robust bioinformatical analysis. In the biological aspect, the work is much more limited. The authors do implore general mechanisms of biology behind covid and Alzheimer’s disease, however they fail to explore the topics in detail and the relation between covid and AD is not fully investigated.

The authors should consider discussing possible mid and long-lasting effects of covid-19 on Alzheimer disease patients, including long-covid or PASC symptoms. Authors should also briefly introduce Alzheimer’s disease causes and symptoms in the introduction, as well as explore the genetic causes leading to variable outcomes of severity of both diseases.

Major points:

The identification of shared biomarkers with high diagnostic potential could have important implications for the development of co-diagnostic tools and the understanding of the interplay between these two complex diseases. I would welcome a clearer explanation of how this work can prove to be valuable in diagnosis and treatment of diseases, as mentioned on lines 158-159. The work could indeed be very useful in this aspect, and therefore there should be more focus placed on it.

On lines 165-179, The work evaluates data from patients. Is there any information on the disease progression of the patients, perhaps even mortality rates? If there is significant prevalence of covid related complications in one of the groups, probably in the Alzheimer patients, could this effect the outcomes or conclusions of the study? For example, if the AD group exhibited a higher level of mortality, could this be caused by exacerbated immune response that this group is prone to?

Lines 245 to 272 discuss the regulation of interaction of metal ions with genes regulated in covid patients. I find this part slightly lacking credibility. Iron, one of the major metal ions figuring in covid infections is not mentioned at all, furthermore it is well known that metal ion homeostasis is important in other infectious diseases as well. Authors should discuss these two points in more detail. Metal ion homeostasis is common in both AD and covid, however this does not mean that there is a direct link between those occurrences, instead it might be rather a symptom of unrelated origin. Authors should consider whether their results could imply that covid could worsen Alzheimer’s disease progression by altering metal ion homeostasis, which would be a intriguing finding if supported by further studies.

Lines 303-304: “after conducting a thorough analysis of immune penetration and metabolic pathways, it was discovered that these diagnostic genes exhibited significant expression levels in various immune cell subtypes and metabolic pathways” is a very strong statement and the analysis and discussion of this topic is only shallow. Therefore, this statement should be altered.

Lines 320-321 state that “During our analysis of AD and COVID-19, we identified a clear correlation in the oxidative phosphorylation pathway among certain core genes” This statement is not clearly supported by the data in the work and should be altered to be explorative, not definitive. Clear in vitro or in vivo data would be needed to fully support this statement, which the authors correctly admit throughout the manuscript.

Minor points:

The study does not address potential factors, such as age, sex, geographic distribution or severity of diseases of the patients. Can the authors provide some insight into this information?

Line 78: the word “infect” should be “infection”?

Line 88-89: The argument that “Mitochondrial dysfunction and oxidative stress contribute to neuronal damage and worsening of Alzheimer's disease” warrants a citation.

Line 256: In vitro should be in italics

Figure 1, A and B don’t bear any significant information value. I would suggest removing these parts of figures, since the data is well described on lines 114-117.

Reviewer #3: Summary:

The question addressed in this manuscript is important in the field of Alzheimer’s disease research since inflammatory insults like COVID-19 infection likely impact innate immunity which appears to become dysregulated in Alzheimer’s disease. The authors try to pin-point what molecular, metabolic or immune biomarkers are shared between COVID-19 infection and Alzheimer’s disease which would further support a possible link between the two conditions. However, they attempt to do this by making an incompatible comparison between microarray RNA sequencing from blood samples of COVID-19 patients with brain samples from Alzheimer’s patients in search for common biomarkers. Their analyses are impressive and could prove to be informative, but the implications of their findings are diminished by the incompatible comparison (and integration of datasets) without any proof that the datasets are indeed comparable (i.e., with a PCA plot to show similarity among samples from different types of biological samples and datasets).

Major comments/suggestions:

1. The integration of the Alzheimer disease datasets is inappropriate since they were sequenced using different microarray platforms that have been reported to be non-comparable (see https://link.springer.com/article/10.1186/1745-6150-3-23). To find DEGs that are differentially regulated in AD using both datasets, it is recommended that a differential gene expression analysis is completed independently for each dataset, then the common down/upregulated genes can be compared to the COVID dataset.

2. Based on the heatmap resulting from their WGCNA analysis in Figure 2C, it appears as though the COVID dataset was combined with the AD datasets for the WGCNA analysis. This is also inappropriate unless the datasets were pre-processed (i.e., individual normalization to sample size factor within each dataset) and including a batch-correction. Since they did not clearly mention of how datasets were normalized or genes were filtered, I assume they combined raw sequence counts for all three datasets for their WGCNA analysis. Furthermore, combining the COVID RNAseq from whole blood with AD RNAseq from various brain regions in this WGCNA analysis is less informative since the gene modules likely reflect tissue-specific expression patterns, not disease-related gene signatures.

3. There is no discussion about how comparing blood versus brain samples could lead to gene overlap (or exclusions) that are not relevant to the diseases they investigate.

4. The authors should have shown if control samples from the different datasets were indeed comparable (i.e., with a PCA plot to show similarity among samples from the three datasets). If the control samples were dissimilar, then one cannot compare/overlap differentially expressed genes between datasets.

Minor comments/suggestions:

1. More information about gene ontology analysis is required since various tools and pipelines have different algorithms for determining enrichment of specific domains.

2. Parameters of the WGCNA pipeline drastically impact the modules so it is important to include exactly what the parameters were (instead of just stating the “optimal” parameter setting was used).

3. Showing the graphical output of intermediate WGCNA steps is unnecessary (and uncommon) but the information from this output (like the optimal soft threshold power that is indicated by the horizontal red line in Figure 2A) should be included in the Methods section.

6. PLOS authors have the option to publish the peer review history of their article (what does this mean?). If published, this will include your full peer review and any attached files.

Reviewer #1: No

Reviewer #2: No

Reviewer #3: No

---

## [Author Response · Author response to Decision Letter 0]

11 Nov 2024

For the first reviewer:

Thank you for your valuable comments and insightful suggestions. We have carefully reviewed each of your points and made the necessary revisions to improve the clarity and quality of our manuscript. Detailed responses to each comment are provided below.

1. Reviewer’s comment: The authors should explain all the abbreviations, DEG for exemple…since the first use

Response: We are grateful to the reviewer for noting the importance of defining abbreviations upon first use, and we have made the suggested adjustments. We have now ensured that all abbreviations, including “DEG” (differentially expressed genes), are defined upon their first use in the manuscript. This greatly improves the clarity of the text for readers unfamiliar with the relevant terminology.

2. Reviewer’s comment: Figure 1, the authors need to precise for panel C &D “common up/down regulated” as maybe common between AD and COVID-19

Response: Thank you for pointing out the clarification needed in Figure 1. We have changed the labels in panel C &D to make it clear that “common up/down regulated” refers to genes that are commonly up- or down-regulated between AD and COVID-19. This adjustment will help avoid any potential ambiguity for the reader. This adjustment will help avoid any potential ambiguity for the reader.

3. Reviewer’s comment: Figure 1 is written in small letter, as a reader we need to use the zoom in, maybe change the size of the police…

Response: Thank you for pointing out that the font is too small. We have modified the relevant font size in the image appropriately, but due to image length issues we are unable to increase it further.

4. Reviewer’s comment: Same remark for figure 2, we can not read without using zoom in

Response: I would like to express my gratitude once more for your invaluable contribution. As with the preceding comment, the font size of the text in Figure 2 has been adjusted in accordance with the requisite specifications.

5. Reviewer’s comment: Figure 2. The authors need to precise which correlation test they used.

Response: I am grateful for your recommendations regarding the methodology for conducting correlation tests. In Figure 2, the Pearson correlation test was employed to analyse the correlation between the two diseases. The legend has been modified to explicitly indicate that the Pearson correlation test was used to assess these correlations.

6. Reviewer’s comment: The authors need to rephrase “the results revealed that these genes were associated with pathways such as…” it is more likely that the pathways involving cytokines etc…

Response: I am grateful for your proposal. In light of the feedback provided, the sentence has been rephrased to clarify that the pathways in question involve biological processes such as cytokines and growth factors, rather than directly associating genes with these molecules. The revised sentence now reads as follows: The results demonstrated that these genes were implicated in pathways associated with cytokines, growth factors, apoptosis, and responses to metal ions and inorganic substances. This amendment enhances the clarity and precision of the biological relationship being delineated. (Lines 146-148)

7. Reviewer’s comment: Figure 5 A, the number on the top of the graph should be explained… represented r? p value ? or other?

Response: Your comments about providing further explanation of the numbers at the top of the picture are much appreciated. To better explain Figure 5 A, we have added the following note to the results section of the main text: Moreover, there is no significant correlation between both nCount RNA and nFeature RNA and percent.mt (correlation coefficients are -0.05 and -0.04, respectively). And the following description has been added to the figure legends: The values at the top of each panel represent the correlation coefficients between the variables shown on the axes. (Lines 249-251)

8. Reviewer’s comment: Figure 6 A and B, the writing on the panels are very small

Response: Thank you again for your suggestion regarding the font size in the image. We have adjusted the text size in Figure 6 to enhance readability.

9. Reviewer’s comment: The authors need to provide the GEO number of AD and COVID19 data set used, in the first sentence of the Materials and Methods/ Data download, the link refered to the principal webpage…and not in the results part

Response: We concur with and are grateful for this recommendation. Consequently, we have amended the wording in the outcome to read: The limma package was utilized to identify differentially expressed genes (DEGs) between the COVID-19 (n = 34) patients and healthy controls (n = 11). And we have incorporated a description of the dataset numbers and URLs in the methods section, as recommended. (Lines 104-105 and 488-493)

10. Reviewer’s comment: The authors need to explain the limma package in short sentence…

Response: We would like to express our sincerest gratitude for this valuable suggestion. In response, we have provided a comprehensive and informative overview of the “limma” package, emphasising its relevance to our analysis. We have also included references to show why we use this package and where readers can find more information. The objective of these additions is to enhance the clarity and rigour of our methodological approach. (Lines 493-496)

11. Reviewer’s comment: The authors should explain their choice for a threshold of 140 to remove the outlier samples.

Response: We agree with this comment, which could make the data processing process more understandable. We have therefore added explanations and, in addition, modified some of the descriptions in order to make the textual descriptions more fluent. (Lines 133-135)

12. Reviewer’s comment: The authors need to explain why they used beta threshold at 6.

Response: We are very grateful for this suggestion. In response, we have provided a more comprehensive explanation of the theoretical basis and rationale behind selecting 6 as the soft threshold power (β). This additional context aims to clarify our choice and enhance the transparency of our methodology. (Lines 511-517)

13. Reviewer’s comment: The authors claimed in legend of the figure 7C co-localisation and in the text, association, maybe they need to explain these 2 concepts…

Response: Thank you for pointing out the potential confusion between the terms "co-localisation" and "association". Indeed, an association between EIF3H and oxidative phosphorylation was obtained by single-cell localisation of immune cells. However, for the sake of rigour and consistent terminology throughout the paper, we have modified the description here and called it "co-localisation". In the discussion section, we further elaborate on the relationship between co-localisation and association. We initially explored the relationship between EIF3H and oxidative phosphorylation by co-localisation analysis of immune cells. This only reflects the association to a certain extent, which needs to be further explored in related studies, so we have improved the presentation here for rigour. We have also revised the presentation of the methods section to make it more in line with academic norms. Thanks again for this great suggestion, which plays an important role in contributing to ensuring rigour and uniformity in the presentation of our articles. (Lines 277-279 , 416-419 and 550-553)

For the second reviewer:

We would like to thank you for your valuable feedback and constructive recommendations. Your comments have been instrumental in improving our work, and we have addressed each point in great detail in the revised manuscript. Please find our detailed responses to your comments below.

1. Reviewer’s comment: The work provides a interesting descriptive statistical analysis of comparing genetic regulation in Alzheimer’s disease and covid-19 patients. The presented statistical pipeline leads into a robust model that could be used for exploring relations between inherited and infectious diseases. The work shows a novel approach and robust bioinformatical analysis. In the biological aspect, the work is much more limited. The authors do implore general mechanisms of biology behind covid and Alzheimer’s disease, however they fail to explore the topics in detail and the relation between covid and AD is not fully investigated. The authors should consider discussing possible mid and long-lasting effects of covid-19 on Alzheimer disease patients, including long-covid or PASC symptoms. Authors should also briefly introduce Alzheimer’s disease causes and symptoms in the introduction, as well as explore the genetic causes leading to variable outcomes of severity of both diseases.

Response: Thank you very much for your acknowledgement of our analytical methods and model construction work, as well as your valuable comments on the biological mechanisms underlying our extended findings. We strongly agree with these important suggestions you made and have revised the Discussion section at greater length to explore the biological significance of the results of this study in greater depth. 

Specifically, we have delved into the biological mechanisms that metal ion homeostasis may play in the pathogenesis and progression of these two diseases. In addition, we have comprehensively elaborated in detail how changes in metal ion levels affect neuroinflammation and oxidative stress, which may play an important role in the interplay between AD and COVID-19, in the context of previously available studies. We have also added consideration of long-term COVID-19 to our discussion of the results, which adds to the length of time uniformity with AD, which is also a chronic course disease, and helps readers to better understand the mechanisms of interaction between these two diseases.

Furthermore, an introduction to the aetiology and symptomatology of Alzheimer's disease has been incorporated into the Introduction section, and an investigation of the genetic factors that contribute to the differing degrees of severity observed in the two diseases has been undertaken. These additions and revisions are intended to enrich the exploration of biological mechanisms and provide a more comprehensive understanding of the potential link between AD and COVID-19.

2. Reviewer’s comment: The identification of shared biomarkers with high diagnostic potential could have important implications for the development of co-diagnostic tools and the understanding of the interplay between these two complex diseases. I would welcome a clearer explanation of how this work can prove to be valuable in diagnosis and treatment of diseases, as mentioned on lines 158-159. The work could indeed be very useful in this aspect, and therefore there should be more focus placed on it.

Response: We are most grateful for the proposal. The discussion of the actual value of these diagnostic biomarkers is really important. We have further explained and illustrated the practical value of these diagnostic biomarkers in the results section. Corresponding to the description of diagnostic biomarkers in the results section, we have added further discussion of their practical implications, such as therapeutic and drug development target selection and the development of diagnostic tools, based on valuable input. (Lines 195-198 and 371-377)

3. Reviewer’s comment: On lines 165-179, The work evaluates data from patients. Is there any information on the disease progression of the patients, perhaps even mortality rates? If there is significant prevalence of covid related complications in one of the groups, probably in the Alzheimer patients, could this effect the outcomes or conclusions of the study? For example, if the AD group exhibited a higher level of mortality, could this be caused by exacerbated immune response that this group is prone to?

Response: This insightful comment is much appreciated. Although this data does not provide data on disease progression or mortality in patients, we do not believe that this will have a significant impact on our main results and conclusions. This is because the focus of our analyses was on core genes and immune infiltration and the correlation between them. These analyses were based on gene expression data. The lack of relevant prognostic data may limit our analyses of the association between disease progression and the strength of the immune response, but our analyses were still able to provide valuable insights into immune cell and gene expression in AD and COVID-19. And we have added relevant notes in the Discussion section to explicitly discuss this data limitation and potential implications. (Lines 453-463)

4. Reviewer’s comment: Lines 245 to 272 discuss the regulation of interaction of metal ions with genes regulated in covid patients. I find this part slightly lacking credibility. Iron, one of the major metal ions figuring in covid infections is not mentioned at all, furthermore it is well known that metal ion homeostasis is important in other infectious diseases as well. Authors should discuss these two points in more detail. Metal ion homeostasis is common in both AD and covid, however this does not mean that there is a direct link between those occurrences, instead it might be rather a symptom of unrelated origin. Authors should consider whether their results could imply that covid could worsen Alzheimer’s disease progression by altering metal ion homeostasis, which would be a intriguing finding if supported by further studies. 

Response: We are grateful to the reviewers for their valuable insights into the regulatory processes governing metal ions, particularly iron ions, in the context of COVID-19 and Alzheimer's disease. We acknowledge that the discussion of this section in the previous version of the manuscript was not sufficiently insightful and appreciate the suggestion that we consider the widespread nature of metal ion homeostasis. 

In response to the reviewers' comments, the discussion section on the physiological and pathological roles of metal ions has been revised and now includes a discussion of the roles of iron, copper, and zinc ions in relation to immunity and oxidative stress. We elaborated on the important role of metal ions, including iron ions, in COVID-19 and AD. 

Furthermore, we have expanded the discussion to emphasise the importance of metal ions, including iron ions, in the context of infectious diseases. We present an insightful discussion of how metal ion dysregulation is a common factor in both diseases and may influence disease course and outcome. Furthermore, we put forth the novel hypothesis that the progression of AD may be influenced by alterations in metal ion homeostasis, which could be a consequence of the effects of SARS-CoV-2 infection. In light of this hypothesis, which offers a novel perspective on the underlying mechanisms, we have revised the manuscript and propose further investigation. It is our contention that this could provide new insights into the pathophysiology of both diseases. 

In addition to the comprehensive and systematic elaboration of the role of metal ions in the aetiology of both SARS-CoV-2 infection and AD in the preceding section of the discussion, we further enhance the possible involvement of metal ions with oxidative stress, or the closely related oxidative phosphorylation, in the co-morbid mechanisms of SARS-CoV-2 infection and AD for further elaboration.

We would like to express our gratitude once more to the reviewers for their insightful comments, which have significantly enhanced the quality and credibility of the manuscript and provided valuable guidance for future research directions. (Lines 314-365)

5. Reviewer’s comment: Lines 303-304: “after conducting a thorough analysis of immune penetration and metabolic pathways, it was discovered that these diagnostic genes exhibited significant expression levels in various immune cell subtypes and metabolic pathways” is a very strong statement and the analysis and discussion of this topic is only shallow. Therefore, this statement should be altered.

Response: I am most grateful for your advice regarding the description of our tone of voice. We concur with your assessment tha

---

## [Decision Letter · Decision Letter 1]

25 Nov 2024

PONE-D-24-36922R1Identification of biomarkers in Alzheimer's disease and COVID-19 by bioinformatics combining single-cell data analysis and machine learning algorithmsPLOS ONE

Dear Dr. Liu,

Thank you for submitting your manuscript to PLOS ONE. After careful consideration, we feel that it has merit but does not fully meet PLOS ONE’s publication criteria as it currently stands. Therefore, we invite you to submit a revised version of the manuscript that addresses the points raised during the review process.

We look forward to receiving your revised manuscript.

Kind regards,

Zhijie Xu

Academic Editor

PLOS ONE

Additional Editor Comments:

The reviewers have provided some important comments. In addition, the emerging reports about COVID-19 should also be discussed, especially like DOI: 10.2174/1573412918666220509032754, doi: 10.14218/JCTH.2022.00339, doi: 10.14218/JCTH.2022.00038, etc.

Reviewers' comments:

Reviewer's Responses to Questions

**Comments to the Author**

1. If the authors have adequately addressed your comments raised in a previous round of review and you feel that this manuscript is now acceptable for publication, you may indicate that here to bypass the “Comments to the Author” section, enter your conflict of interest statement in the “Confidential to Editor” section, and submit your "Accept" recommendation.

Reviewer #2: All comments have been addressed

Reviewer #3: (No Response)

2. Is the manuscript technically sound, and do the data support the conclusions?

Reviewer #2: Yes

Reviewer #3: Partly

3. Has the statistical analysis been performed appropriately and rigorously? 

Reviewer #2: Yes

Reviewer #3: I Don't Know

4. Have the authors made all data underlying the findings in their manuscript fully available?

Reviewer #2: Yes

Reviewer #3: (No Response)

5. Is the manuscript presented in an intelligible fashion and written in standard English?

Reviewer #2: Yes

Reviewer #3: Yes

6. Review Comments to the Author

Reviewer #2: Minor correction: In several places of the revised manuscript, there is a repeated error in the word "COIVD". Should Be COVID. Lines 196, 458, 460, 474.

The authors addressed my comments and modified the manuscript accordingly. Although I still find some of the discussion regarding biological mechanisms limited, as the authors correctly state, that is not the main focus of the presented work.

In general I find the work robust, exploring a novel and interesting area with a positive impact to the scientific community.

After correcting grammatical errors, I recommend the work for publication.

Reviewer #3: After another review of the revised manuscript, the other reviewers’ comments, and the authors’ response to all reviewers, I think the manuscript needs another round of revisions to be suitable for publication. The major critiques below largely stem from the lack of detail regarding the integration of three bulk RNA sequencing datasets and their other bioinformatic methods (see methodological details requested in major concerns 1, 3-6, 8). The authors’ response to my initial comments includes more detail about their data handling methods, but these details need to be present in the manuscript. Although the authors’ methods may be “…reasonable and rigorous,” the authors need to provide enough detail about their bioinformatics pipeline(s) to allow readers to easily understand the reliability or implications of the data.

Minor -

1. It is unclear what samples from each dataset were used. The AD datasets were combined to result in 232 Alzheimer's patients and 178 healthy controls, but there are more than 410 cumulative samples in those datasets so a subset of samples must have been selected.

2. Typos remain throughout the manuscript.

Major -

1. Although the authors mention that the limma package “…has powerful read, normalization, and data exploration capabilities,” they should include more precise methodology regarding the normalization of and batch-correction between the datasets. I suggest they add in the details they outline in their response to Reviewer #3’s third and fifth comment into their methods.

2. They should include discussion of how the biomarkers they identify overlap/compare with previous studies (and cite those studies) as they suggest they do in the response (3A) to reviewer #3’s third comment.

3. The methods and datasets used to calculate the positive predictive value of these eight identified genes are missing completely. AUC is undefined. Were the ROC curves based on blood samples only (these details need to be included in the methods)? If not, then I suggest re-running these analyses on blood samples only to claim that these genes are potential diagnostic markers. Also, it appears as though the ROC curves are based on presence/absence of COVID and AD (is that correct?). The authors should show the predictability of the eight genes on each disease alone since it would provide insight into whether some genes are more predictive of AD or COVID diagnosis, or if they are specific to the co-diagnosis.

4. Was the AD immune infiltration analysis conducted on composite (blood+brain) dataset? If so, was the analysis on done on the ComBat normalized composite dataset? I ask because I imagine cross-tissue normalization could impact the ability/accuracy of a pipeline like CIBERSORT to determine cell-types within those normalized datasets. In the methods, the authors should add in what datasets were used and how data was handled prior to the CIBERSORT analysis.

5. Although the authors have revised the discussion to explicitly state that there are limitations due to cross-tissue analyses, I do not agree with the authors in that they have provided “a more comprehensive analysis of the limitations and potential implications of the data…” since they only mention:

521-525: “Furthermore, the cross-tissue nature of the analysis of AD derived from brain tissue versus that of the analysis of COIVD-19 [**typo error] derived from blood may potentially impact the accuracy of the results due to tissue-specific factors. However, we endeavoured to minimise the influence of tissue specificity through the utilisation of the aforementioned methods.”

Only stating that they “endeavoured to minimise the influence of tissue specificity…” is insufficient in The limitations should be discussed in more detail to include discussion of potential non-disease-specific influences (i.e., due to their cross-tissue integration) and why their main findings are likely disease- and not tissue-specific. Additionally, the implications of their findings would be bolstered by discussion of whether (or more so why) they think altered expression of the eight identified genes reflect brain-specific, blood cell-specific or shared mechanisms between tissues (based on what is known tissue-specific gene expression and cell-type-specific functions of their eight identified genes; some of their ideas are included in the response to comment #3). Lastly, the methodology used to minimize tissue-specific influences is not detailed (what are the “aforementioned methods” [is it ComBat?] and is there prior proof that they minimize tissue-specific effects thus allowing for identification of trait-relevant mechanisms?).

6. At line 120, the authors have removed details about the bulk RNA sequencing data that was analyzed (removed: “…we acquired and integrated the Alzheimer's disease datasets GSE5281 and GSE63060 (based on platforms GPL570 and GPL21185, respectively)” and replaced it with “…we acquired and integrated two AD datasets”) which decreases the detail necessary to understand that two RNA-seq datasets from two different tissues (brain and blood) that were acquired using different platforms were integrated then analyzed as a single dataset. After further review of current methods for this type of integration, I agree with the authors in that this type of integration can be done and prove to be informative. However, the integration of datasets is complex and non-trivial, so the authors need to detail their integration method which remains missing in the Methods section (i.e., no mention of the ComBat approach). Furthermore, the authors do not perform brain-specific analyses, so it is not apparent why they needed to combine a brain and blood AD dataset to begin with.

7. The authors do however did satisfy the suggestion to show that the datasets can be well-integrated for the WGCNA (with the PCA plots in Supp Fig 1), but they did not provide proof of the equivalent analysis for the integration of the two AD datasets alone (which was done for the DEG analysis).

8. ‘ssGSEA’ is not defined and the methods they used for determining enrichment via ssGSEA are missing. Are the scale bars/heatmap colors for Fig 4E & 4F representing the correlation coefficient? The authors should explicitly state this in the figure legend. Also, in Figure 4’s legend, it says “Correlation between the expression levels of seven [**Typo? Should be eight?**] hub genes and the ssGSEA enrichment scores…”.

9. Since the authors are arguing that the differential expression of the eight identified genes are shared between COVID and AD, the authors should reorganize the Figure 4B, 4D, 4E and 4F so that the genes and pathways are in the same order in each figure panel to allow readers to more readily evaluate the similarity of gene and immune cell abundance or pathway correlations in AD and COVID.

7. PLOS authors have the option to publish the peer review history of their article (what does this mean?). If published, this will include your full peer review and any attached files.

Reviewer #2: No

Reviewer #3: No

---

## [Author Response · Author response to Decision Letter 1]

29 Dec 2024

Dear Editor,

I am grateful for your meticulous examination and invaluable counsel regarding our latest revised manuscript. Your expert feedback has been instrumental in further enhancing the manuscript. On behalf of our research team, I would like to express our sincerest gratitude to you. All changes to the previous version of the manuscript, including additions, deletions and modifications, are indicated by red text. In this response, the number of lines is indicated according to the document ‘Revised Manuscript with Track Changes’.

We have firstly revised “the emerging reports about COVID-19” based on your valuable comments. The discussion and citation of the emerging reports about COVID-19 have been incorporated into the preface and discussion sections. 

Furthermore, the following three papers, which we consider to be of particular note, have been referenced in lines 61, 465, and 106.

DOI: 10.2174/1573412918666220509032754

DOI: 10.14218/JCTH.2022.00339

DOI: 10.14218/JCTH.2022.00038

With kind regards, 

Yours sincerely, 

Juntu Li

Jun Liu

Department of Critical Care Medicine and Emergency, The Affiliated Suzhou Hospital of Nanjing Medical University (Suzhou Municipal Hospital), Gusu School, Nanjing Medical University, Suzhou Clinical Medical Center of Critical Care Medicine,

No. 16, West Baita Road, Suzhou, Jiangsu Province, China 215001

E-mail: liujunphd@sina.cn

The following section presents a comprehensive response to each of the reviewers' queries.

For the second reviewer:

We sincerely appreciate your valuable feedback. We have carefully addressed all your comments and revised our manuscript accordingly. These changes have substantially improved the quality of our work. Below, we provide point-by-point responses to your comments.

1. Reviewer’s comment: Minor correction: In several places of the revised manuscript, there is a repeated error in the word "COIVD". Should Be COVID. Lines 196, 458, 460, 474. 

The authors addressed my comments and modified the manuscript accordingly. Although I still find some of the discussion regarding biological mechanisms limited, as the authors correctly state, that is not the main focus of the presented work.

In general I find the work robust, exploring a novel and interesting area with a positive impact to the scientific community.

After correcting grammatical errors, I recommend the work for publication.

Response: We would like to express our gratitude for your attention to detail in correcting the spelling of certain words in our previous version of the manuscript. The requisite corrections have been made (lines 207, 505, 508, and 521), and the vocabulary has been meticulously examined in order to guarantee the correct spelling.

We appreciate your continued support in improving this manuscript.

For the third reviewer:

We greatly appreciate your thorough review and constructive feedback regarding our methodology. In response, we have substantially expanded the Methods section to provide comprehensive details about our analytical pipeline. Please find below a comprehensive response to each of your suggestions.

1. Reviewer’s comment: It is unclear what samples from each dataset were used. The AD datasets were combined to result in 232 Alzheimer's patients and 178 healthy controls, but there are more than 410 cumulative samples in those datasets so a subset of samples must have been selected.

Response: I am grateful for your interest in the topic of sample size following the merging of our datasets. The AD patient group from dataset GSE5281 comprises 87 samples, while the healthy control group comprises 74 samples. Dataset GSE63060 includes 145 samples from the AD patient group, 104 samples from the healthy control group, and 80 samples from the Mild Cognitive Impairment (MCI) patient group. The complete AD patient and healthy control groups from the aforementioned datasets were included in the analysis. As MCI is not a component of AD, it was not included in our study. Furthermore, a comprehensive account of the methodology employed is provided in the Methods section of the article (lines 560-562), with the aim of offering the reader a more nuanced comprehension of the research data utilised.

2. Reviewer’s comment: Typos remain throughout the manuscript.

Response: I would like to express my gratitude for your attention to the spelling issue in our manuscript. The manuscript has been subjected to a rigorous spell-checking process, resulting in the correction of some errors, including the replacement of the erroneous “COIVD-19” with the correct “COVID-19” (lines 207, 505, 508, and 521). This makes the article more rigorous and brings it in line with the standard requirements for publication in this journal.

3. Reviewer’s comment: Although the authors mention that the limma package “…has powerful read, normalization, and data exploration capabilities,” they should include more precise methodology regarding the normalization of and batch-correction between the datasets. I suggest they add in the details they outline in their response to Reviewer #3’s third and fifth comment into their methods.

Response: Thank you for this valuable suggestion to include a more accurate method for normalising and batch correcting datasets. We have already specified this in the previous version of the response letter. We very much agree to include a detailed description of this in the methodology section of the manuscript. We have condensed and distilled it and included it in the main text in a form more suitable for description in the text (lines 577-584).

4. Reviewer’s comment: They should include discussion of how the biomarkers they identify overlap/compare with previous studies (and cite those studies) as they suggest they do in the response (3A) to reviewer #3’s third comment.

Response: Thank you for your valuable advice. We have partially discussed the overlap of these biomarkers with previous studies in the discussion section of the previous manuscript. Based on your constructive comments, we have further explored in depth the overlap of certain biomarkers with previous relevant studies. In particular, we are notable for BCL6, a gene involved in the pathogenesis of both diseases we have studied (lines 401-415). And appropriate citations have been added. This contributes significantly to the richness of our discussion section.

5. Reviewer’s comment: The methods and datasets used to calculate the positive predictive value of these eight identified genes are missing completely. AUC is undefined. Were the ROC curves based on blood samples only (these details need to be included in the methods)? If not, then I suggest re-running these analyses on blood samples only to claim that these genes are potential diagnostic markers. Also, it appears as though the ROC curves are based on presence/absence of COVID and AD (is that correct?). The authors should show the predictability of the eight genes on each disease alone since it would provide insight into whether some genes are more predictive of AD or COVID diagnosis, or if they are specific to the co-diagnosis.

Response: Thank you for your suggestions regarding our ROC analysis. We appreciate your careful review and would like to clarify several points:

First, we confirm that our ROC curves were indeed based on blood samples only, considering the technical variations and tissue heterogeneity. We have now revised the methods section to provide more detailed description of our analysis approach (lines 615-618).

Regarding your suggestion about analyzing the predictive power of these genes for each disease separately, we appreciate this perspective. However, our study specifically aimed to investigate the shared molecular signatures between AD and COVID-19, for several key reasons:

1) These eight genes were identified through an integrated analysis pipeline (combining differential expression analysis, WGCNA, and machine learning methods) specifically designed to detect shared molecular features and common pathways between the two diseases. Therefore, our ROC analysis was intentionally focused on validating their collective predictive power for these shared pathological processes.

2) Growing evidence suggests potential molecular links between COVID-19 and AD. Our study aimed to investigate these specific connections, using ROC analysis to validate these shared molecular signatures rather than identifying disease-specific diagnostic markers. This approach is consistent with the findings of recent studies that suggest a potential influence of COVID-19 on the molecular pathology of AD[1].

3) The high AUC values we obtained demonstrate the robustness of these genes as indicators of shared pathological processes between these conditions. This finding supports our research objective of understanding the molecular intersection between AD and COVID-19, which could be particularly valuable for understanding the neurological implications of COVID-19 in the context of AD.

6. Reviewer’s comment: Was the AD immune infiltration analysis conducted on composite (blood+brain) dataset? If so, was the analysis on done on the ComBat normalized composite dataset? I ask because I imagine cross-tissue normalization could impact the ability/accuracy of a pipeline like CIBERSORT to determine cell-types within those normalized datasets. In the methods, the authors should add in what datasets were used and how data was handled prior to the CIBERSORT analysis.

Response: Expressions of gratitude are extended for the valuable suggestions provided. The CIBERSORT analysis is grounded in a standardised dataset, a practice that ensures the comparability of samples and maintains their biological relevance. In light of the suggestions, deliberations have been held regarding the potential ramifications of data standardisation on ensuing analyses.

Firstly, the CIBERSORT algorithm employs the LM22 feature matrix, which is utilised for immune cell deconvolution. The LM22 contains expression features for 22 immune cell types, including T-cells, B-cells, NK-cells, and so forth, and is a well-proven reference[2]. Irrespective of the data preprocessing method employed, this standardised reference matrix enables accurate immune cell proportion estimation[3]. The rationale behind this is twofold. Firstly, the algorithm compares relative gene expression values between the sample and the reference matrix, rather than absolute values. Secondly, the algorithm is implemented in a way that focuses on the expression characteristics of the immune cell types that are present in our standardised processed data. This is advantageous because the data we used has a clear distinction between disease and control groups.

While acknowledging the value of tissue-specific analyses as a analytical strategy, our integrative approach offers enhanced access to the comprehensive array of immune pattern alterations in cases of COVID-19 compared to AD. This is a crucial aspect for initial exploration of the link between the two in terms of pathological mechanisms. In response to the recommendations, we have expanded the Methods section of the manuscript to include a detailed description of the utilized datasets and the employed data preprocessing methods (lines 628-632). This has significantly enhanced the clarity of the methods section of the article.

7. Reviewer’s comment: Although the authors have revised the discussion to explicitly state that there are limitations due to cross-tissue analyses, I do not agree with the authors in that they have provided “a more comprehensive analysis of the limitations and potential implications of the data…” since they only mention:

521-525: “Furthermore, the cross-tissue nature of the analysis of AD derived from brain tissue versus that of the analysis of COIVD-19 [**typo error] derived from blood may potentially impact the accuracy of the results due to tissue-specific factors. However, we endeavoured to minimise the influence of tissue specificity through the utilisation of the aforementioned methods.”

Only stating that they “endeavoured to minimise the influence of tissue specificity…” is insufficient in The limitations should be discussed in more detail to include discussion of potential non-disease-specific influences (i.e., due to their cross-tissue integration) and why their main findings are likely disease- and not tissue-specific. Additionally, the implications of their findings would be bolstered by discussion of whether (or more so why) they think altered expression of the eight identified genes reflect brain-specific, blood cell-specific or shared mechanisms between tissues (based on what is known tissue-specific gene expression and cell-type-specific functions of their eight identified genes; some of their ideas are included in the response to comment #3). Lastly, the methodology used to minimize tissue-specific influences is not detailed (what are the “aforementioned methods” [is it ComBat?] and is there prior proof that they minimize tissue-specific effects thus allowing for identification of trait-relevant mechanisms?).

Response: We would like to express our sincere gratitude for the insightful suggestions you have provided with regard to the tissue specificity and function of the genes outlined in our manuscript. It is acknowledged that the preceding version of the manuscript lacked the requisite level of detail, particularly in addressing the cross-tissue analysis limitations and the potential implications of tissue-specific factors. In light of the constructive feedback provided, substantial revisions have been made to comprehensively address the concerns raised. Specifically, the following changes have been made:

1) Enhanced the discussion of cross-tissue analysis validity from biological perspectives, including evidence of blood-brain barrier permeability and shared inflammatory markers between CNS and blood in both diseases (lines 524-531). 

2) Clarified our methodological approach, particularly the use of ComBat method to eliminate tissue batch effects, and explained how gene chip technology ensures sample comparability (lines 531-539).

3) Provided an in-depth analysis of BCL6 as a key example, demonstrating how its involvement in both diseases through conserved molecular pathways represents true disease mechanisms rather than tissue-specific effects (lines 401-415). 

In this revision, a detailed discussion on the sharing of genes between different tissues has also been provided. This discussion is supported by relevant literature which demonstrates the biological basis for cross-tissue biomarker studies. The content has undergone a thorough revision process that involved the integration of select components from the prior response to the reviewers. In addition, the description has been revised in the light of the update and the "method" mentioned in the previous manuscript has been clarified in the description.

The incorporation of constructive feedback, our deliberations, and the revision of this section of the content have culminated in a more lucid description of the manuscript, a more cogent discussion, and an enhanced credibility of the results. It is our conviction that these revisions have considerably strengthened the scientific rigor of our cross-tissue analysis.

8. Reviewer’s comment: At line 120, the authors have removed details about the bulk RNA sequencing data that was analyzed (removed: “…we acquired and integrated the Alzheimer's disease datasets GSE5281 and GSE63060 (based on platforms GPL570 and GPL21185, respectively)” and replaced it with “…we acquired and integrated two AD datasets”) which decreases the detail necessary to understand that two RNA-seq datasets from two different tissues (brain and blood) that were acquired using different platforms were integrated then analyzed as a single dataset. After further review of current methods for this type of integration, I agree with the authors in that this type of integration can be done and prove to be informative. However, the integration of datasets is complex and non-trivial, so the authors need to detail their integration method which remains missing in the Methods section (i.e., no menti

---

## [Editor Report · Decision Letter 2]

8 Jan 2025

Identification of biomarkers in Alzheimer's disease and COVID-19 by bioinformatics combining single-cell data analysis and machine learning algorithms

PONE-D-24-36922R2

Dear Dr. Liu,

We’re pleased to inform you that your manuscript has been judged scientifically suitable for publication and will be formally accepted for publication once it meets all outstanding technical requirements.

Kind regards,

Zhijie Xu

Academic Editor

PLOS ONE

Additional Editor Comments (optional):

The quality of this revised manuscript has significantly improved. The author has carefully addressed the reviewers' questions. This revised manuscript can be accepted for publication.
---

## [Editor Report · Acceptance letter]

16 Jan 2025

PONE-D-24-36922R2 

PLOS ONE

Dear Dr. Liu, 

I'm pleased to inform you that your manuscript has been deemed suitable for publication in PLOS ONE. Congratulations! Your manuscript is now being handed over to our production team.

Kind regards, 

on behalf of

Prof. Zhijie Xu 

Academic Editor

PLOS ONE